# Structural basis of specific lysine transport by *Pseudomonas aeruginosa* permease LysP

Deniz Bicer[1,2,3,4,15,16], Rei Matsuoka [5,16], Aurélien F. A. Moumbock [1,2,3,6],
Preethi Sukumar[7], Albert Suades[8], Harish Cheruvara[9], Andrew Quigley [9],
David Drew[8], Els Pardon [10,11], Jan Steyaert [10,11], Peter J. F. Henderson [7],
Martin Caffrey [12], Julia J. Griese [4] & Emmanuel Nji [1,2,3,12,13,14] ✉

Under conditions of extreme acidity, the lysine-specific permease, LysP, not only mediates the import of L-lysine it also interacts with the transcriptional regulator, CadC, to activate expression of the *cadAB* operon. This operon encodes the lysine decarboxylase, CadA, which converts lysine to cadaverine while consuming a cytoplasmic proton, and the antiporter, CadB, which exports protonated cadaverine in exchange for extracellular lysine. Together, these processes contribute to cytoplasmic pH homeostasis and support bacterial acid resistance - a mechanism essential for the survival of pathogenic bacteria in acidic host environments. Here, we present the cryo-EM structure of LysP from *Pseudomonas aeruginosa* in an inward-occluded conformation (3.2–5.3 Å resolution), bound to L-lysine and a nanobody. L-Lysine is coordinated by hydrophobic contacts, cation–π interactions, and by hydrogen bonding mostly with polar uncharged residues. Reconstitution of LysP into proteoliposomes confirms specific L-lysine transport, which is competitively inhibited by L-4-thialysine. These findings provide a structural framework for understanding selective lysine recognition and inhibition, with implications for antibacterial drug design.

Antibiotic resistance is a significant global public health threat that can have severe consequences, including increased morbidity, mortality, and healthcare costs[1–4]. The number of deaths due to antibiotic-resistant bacteria has been increasing in the European Union and the European Economic Area, with over 25,000 deaths reported in 2007[5], 33,000 in 2015[5] and 38,710 in 2019[6]. The increase in deaths in 2019 was associated with an estimated 865,767 infections of selected antibiotic-resistant bacteria, leading to 1,101,288 total disability-adjusted life-years (DALYs)[6]. In the United States, antibiotic resistance is also a significant public health threat, with more than 2.8 million people infected and over 35,000 deaths reported each year due to antibiotic-resistant infections[7]. Low- and middle-income countries, including many in Africa, face even greater challenges[3,4,8]. Limited healthcare infrastructure, lack of access to quality medical care, and the

[1]BioStruct-Africa, Nairobi, Kenya. [2]BioStruct-Africa, Kumasi, Ghana. [3]BioStruct-Africa, Stockholm, Sweden. [4]Department of Cell and Molecular Biology, Uppsala University, Uppsala, Sweden. [5]OMass Therapeutics Ltd, Building 4000, Chancellor Court, John Smith Drive, ARC Oxford, Oxford, UK. [6]Institute of Pharmaceutical Sciences, Albert-Ludwigs-Universität Freiburg, Freiburg, Germany. [7]Astbury Centre for Structural Molecular Biology, University of Leeds, Leeds, UK. [8]Department of Biochemistry and Biophysics, Science for Life Laboratory, Stockholm University, Stockholm, Sweden. [9]Membrane Protein Laboratory, Diamond Light Source Ltd., Research Complex at Harwell, Didcot, UK. [10]VIB-VUB Center for Structural Biology, Brussels, Belgium. [11]Structural Biology Brussels, Vrije Universiteit Brussel, Brussels, Belgium. [12]Schools of Medicine and Biochemistry & Immunology, Trinity College, Dublin, Ireland. [13]Department of Parasitology and Microbiology, Centre for Research in Infectious Diseases, Yaoundé, Cameroon. [14]Visiting Research Fellow, Membrane Protein Laboratory, Diamond Light Source Ltd., Research Complex at Harwell, Didcot, UK. [15]Present address: Department of Molecular Biology and Genetics, Aarhus University, Aarhus, Denmark. [16]These authors contributed equally: Deniz Bicer, Rei Matsuoka. ✉e-mail: emmanuel.nji@biostructafrica.org

widespread use of antibiotics without proper oversight exacerbate the situation[8]. Consequently, these regions bear a disproportionate burden of antibiotic-resistant infections, resulting in higher morbidity and mortality rates[8]. Furthermore, according to the World Health Organization (WHO), if appropriate measures are not taken to address antibiotic resistance, the number of deaths due to antibiotic-resistant infections is predicted to rise significantly, with a projected 10 million deaths per year globally by 2050[9]. This threat further emphasizes the importance of developing strategies to address antibiotic resistance, such as promoting the appropriate use of antibiotics in medicine, veterinary and agriculture, promoting the use and development of vaccines and tools for early diagnosis of bacterial infections, educating the community about the danger of antibiotic resistance, and investing in research to find alternative treatments for bacterial infections.

Here, we investigate the structure-function relationship of the lysine-specific permease (LysP), a protein involved in the transport of lysine, an essential amino acid, across the inner cell membrane in *P. aeruginosa*. Under conditions of extremely low pH, as in the mammalian stomach, LysP plays a crucial role in the survival of bacteria by interacting with the transcriptional regulator CadC to upregulate the expression of the *cadBA* operon[10–12]. The *cadBA* operon encodes two proteins: CadA, a lysine decarboxylase that converts lysine to cadaverine while consuming a cytoplasmic proton, and CadB, an antiporter that exports cadaverine in exchange for extracellular lysine (Fig. 1a)[10–14]. This decarboxylation reaction not only helps neutralize acidity by consuming protons but also contributes to maintaining the proton gradient that drives cadaverine export, thereby supporting acid resistance in bacteria[15].

This process is important for bacterial survival in the host, as it allows the bacteria to maintain a neutral pH environment, which is necessary for growth[10–12]. The LysP protein from *Pseudomonas aeruginosa* is similar in sequence to LysP from other enteric pathogens (Fig. 1b, c), which cause a significant disease burden worldwide, contributing to more than 96 million DALYs globally[16].

The structure and mechanism of LysP from *P. aeruginosa* (hereafter referred to as LysP), a pathogenic bacterium, is of great interest as this organism is on the WHO list of priority pathogens, known as ESKAPE (*Enterococcus faecium*, *Staphylococcus aureus*, *Klebsiella pneumoniae*, *Acinetobacter baumannii*, *Pseudomonas aeruginosa*, and *Enterobacter species*), for research and development of new antibiotics[17].

*P. aeruginosa* can cause infections in the urinary tract, respiratory system, soft tissue, bones and joints, and many systemic infections such as bacteremia and dermatitis, particularly in hospitalized patients with cystic fibrosis, severe burns, cancer and AIDS patients whose immune systems have been compromised[18–20]. Indeed, 10% of all hospital-acquired infections are caused by *P. aeruginosa* which is difficult to treat because of a naturally high antibiotic resistance. These patients have a mortality rate as high as 40%[18–21].

LysP is an integral membrane lysine specific transporter that belongs to the amino acid, polyamine and organocation (APC) transporter superfamily[22,23]. APC family transporters are present in all kingdoms of life where they play important roles in cell physiology and in disease. They are involved in nutrient uptake, elimination of toxic waste, and exchange of information and signals[22,23]. The human solute carriers (SLC) are part of the APC transport family.

While the crystal and/or cryo-EM structures of ApcT (Na⁺-independent amino acid transporter)[23], AdiC (arginine-agmatine antiporter)[24–26], GadC (glutamate-GABA antiporter)[27], AgcS (Na⁺-alanine symporter)[28], BasC (alanine-serine-cysteine exchanger)[29], CCC (cation-chloride cotransporter)[30–37], and KimA (K⁺/H⁺-symporter)[38], along with decades of biochemical and biophysical studies have provided valuable insights into the selectivity and mode of action of these transporters, there is still much to learn about the structure-function relationship of this diverse family of transporters.

For example, many amino acid transporters are known to exhibit promiscuous substrate recognition, meaning that they can transport multiple substrates with varying affinities[23,39]. The structural basis for this promiscuity is not well understood, and further research is necessary to elucidate the mechanisms of substrate recognition and transport.

Here, we investigate a transporter that specifically transports lysine from *P. aeruginosa* using single-particle cryo-EM and functional approaches. The structure of LysP bound to L-lysine and in complex with a nanobody, as well as accompanying functional data and normal mode analysis reveals useful molecular insights into how LysP, unlike the broad-spectrum amino acid transporters, utilizes a proton gradient to transport L-lysine with high specificity, and how LysP is inhibited by S-(2-aminoethyl)-L-cysteine (L-4-thialysine), which opens up avenues for structure-based antibiotics design to target bacterial LysP.

## Results

### LysP specifically binds and transports L-lysine using a proton gradient

To establish whether LysP is a lysine-specific transporter, LysP was reconstituted into liposomes and the transport of tritiated (³H) L-lysine was measured in the presence of an excess of the twenty cold proteinogenic amino acids as well as L-4-thialysine and D-lysine. The robust transport of ³H-[L-lysine] (Fig. 2a) was inhibited only by cold L-lysine and L-4-thialysine and not by the other amino acids tested, indicating that LysP is a lysine-specific transporter (Fig. 2b, c). The transport inhibition data performed in liposomes agrees with binding data obtained by microscale thermophoresis (MST). The $K_d$ for L- and D-lysine was 14 and 151 μM, respectively (Fig. 2d, Supplementary Fig. 1a, b) and for L-4-thialysine, 31 μM (Fig. 2e). For L-arginine, the $K_d$ for binding to LysP was 15 mM (Supplementary Fig. 1c), and in addition, the IC$_{50}$ for inhibition of [³H]-L-lysine transport by L-arginine was 1.4 mM while that of cold L-lysine was 2 μM (Fig. 2b). These values explain why no L-arginine binding was detected by MST across the concentration range used for L-lysine (Fig. 2d). Similarly, no detectable binding was observed within the concentration range used for L-lysine for the other amino acids tested by MST (Supplementary Fig. 1d and Supplementary Fig 2a–j). Finally, proteoliposomes reconstituted with a proton gradient (pH 7 inside and pH 4 outside) showed robust transport for L-lysine (Fig. 2f). No significant transport was observed with proteoliposomes reconstituted either without a proton gradient (pH 4 inside and pH 4 outside) (Fig. 2f) or with the addition of a sodium gradient (50 mM NaCl outside and 0 mM inside) (Fig. 2g). These results demonstrate that LysP uses a proton gradient for L-lysine transport.

### Cryo-EM structure determination

Several attempts to obtain well-diffracting crystals of LysP failed[40,41]. To obtain stable protein for structure determination, camelid nanobodies were generated and screened against LysP (Supplementary Fig. 3a, b). The best binder Nb5755 improved the stability of LysP with an increase in melting temperature of 20 °C (Supplementary Fig. 4a) and the LysP-L-lysine-Nb5755 complex was used for cryo-EM studies. The binding of Nb5755 to LysP was also evident as a peak shift during size exclusion chromatography (SEC) towards higher molecular weight (Supplementary Fig. 4b), as well as analysis on SDS PAGE of the SEC peak fraction of the complex which showed two bands corresponding to LysP and Nb5755 (Supplementary Fig. 4c). The LysP protein without nanobody in solution revealed predominantly an α-helical profile as judged by circular dichroism (Supplementary Fig. 4d), and importantly, LysP was functional in transport assays (Fig. 2a), making the sample suitable for single-particle cryo-EM analysis. Particle averaging yielded a cryo-EM map with an overall resolution of ~3.68 Å for the LysP-L-lysine-Nb5755 complex (Fig. 3a–c, Table 1, Supplementary Figs. 4e, 5a–g).

                                                         

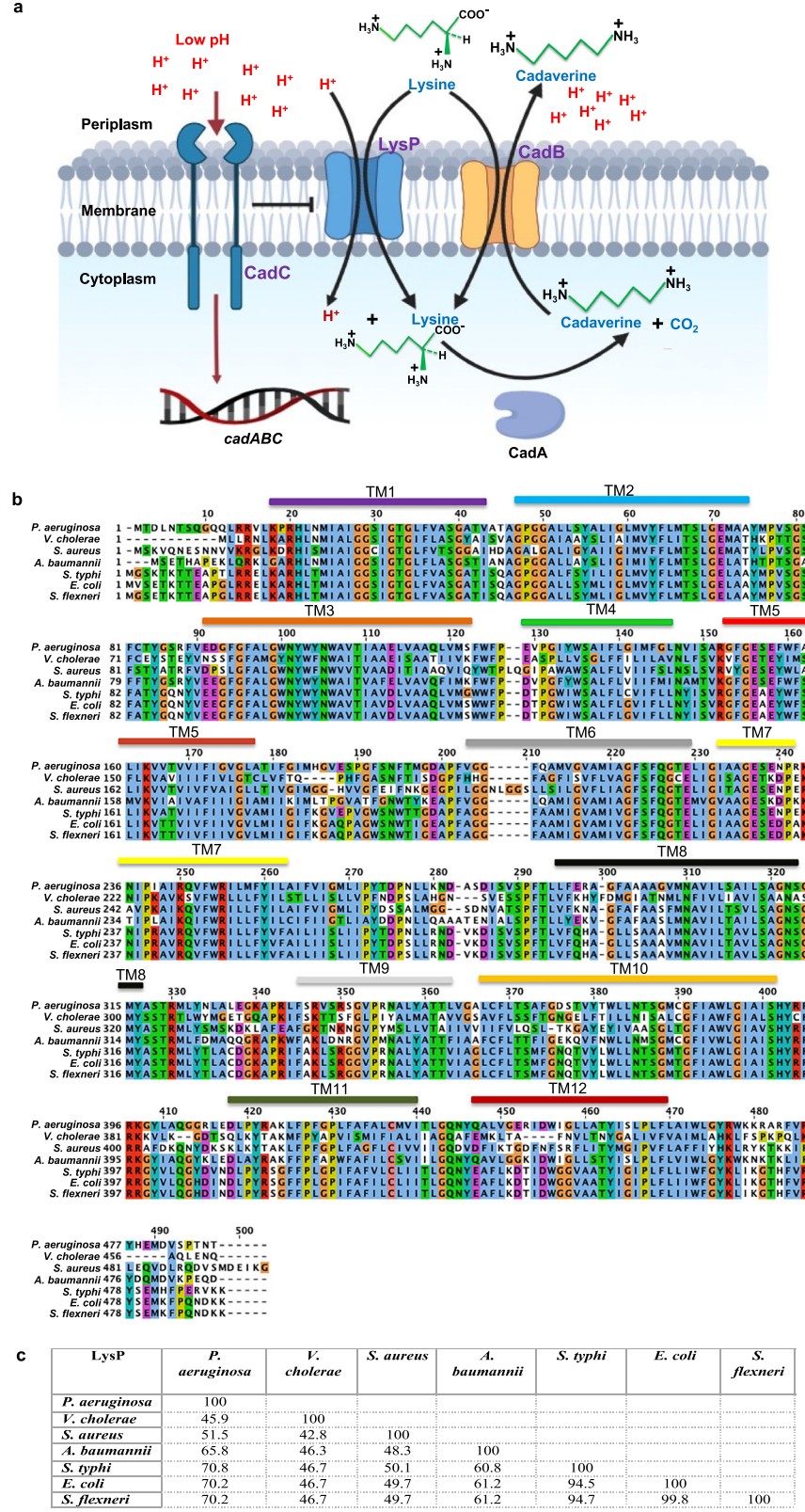

**Fig. 1 | Proposed L-lysine-specific transport mechanism and amino acid sequence analysis of LysP from different bacterial species. a** Schematic representation of bacterial lysine-specific transport mechanism and its role in extremely low pH regulation. Created in BioRender. Lab, D. (2025) https://BioRender.com/p6t92fm. **b** Sequence alignment of *P. aeruginosa* LysP and other enteric pathogenic bacteria (*Vibrio cholerae, Staphylococcus aureus, Acinetobacter baumannii, Salmonella typhi, Escherichia coli,* and *Shigella flexneri*). **c** Number of matching residues (percentage identity) in the alignments shown in **b**.

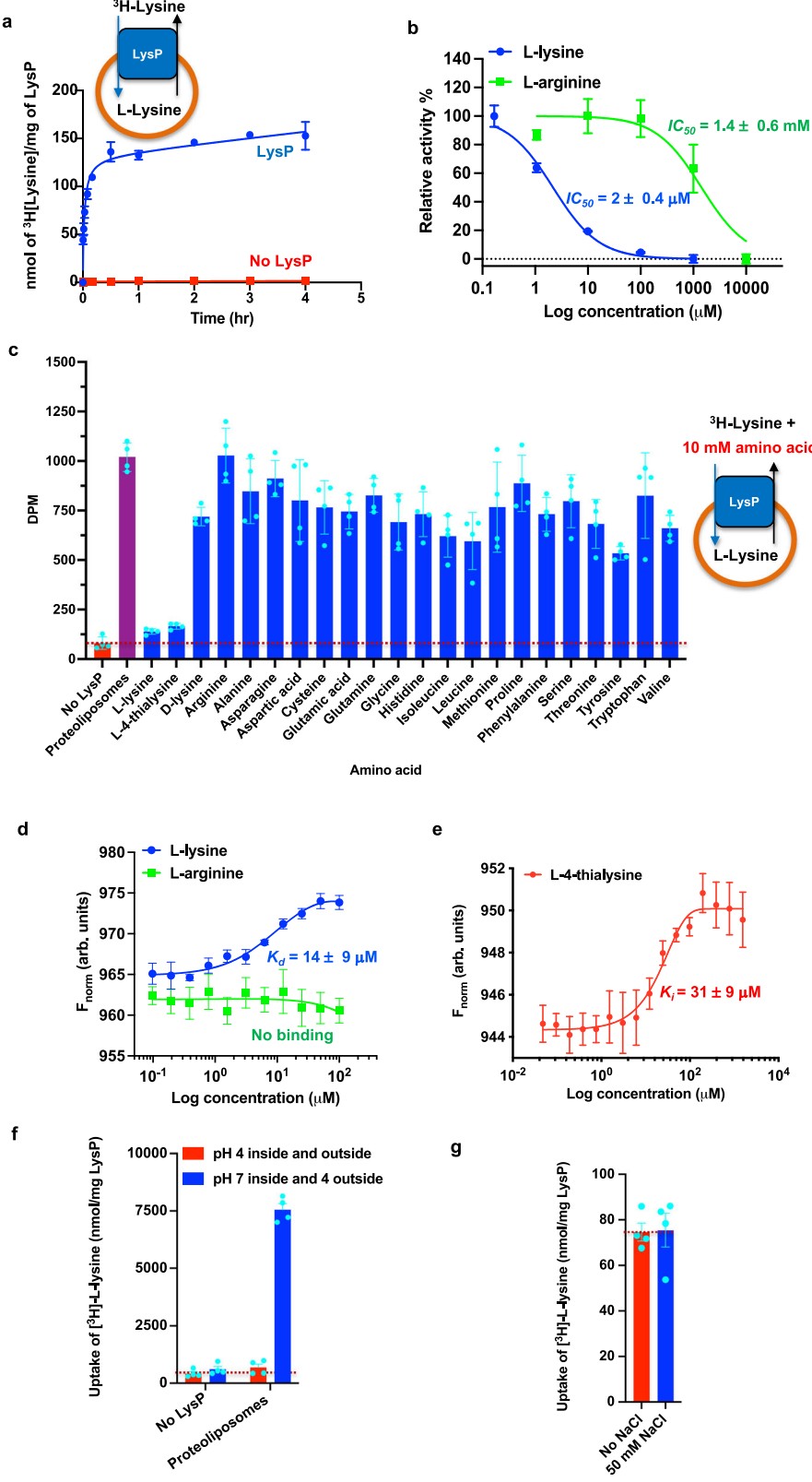

## Structure of LysP–L-Lysine–Nb5755 in an inward-occluded state

The structure of the LysP-nanobody complex was determined in the presence of L-lysine (Fig. 3c, d). LysP adopted an inward-occluded state in the transport cycle with the nanobody binding to LysP on its periplasmic side (Fig. 4a). Superimposing the AlphaFold2-predicted outward-facing structure of LysP with our experimental structure further supports the view that LysP adopts an inward-occluded conformation

(Fig. 4a). Unlike the AlphaFold2 structure where transmembrane helices TM1a and TM6b are in a closed configuration, in the LysP-nanobody complex structure, TM1a and TM6b have opened, in preparation for releasing lysine into the cytosol (Fig. 4a).

The LysP structure reveals 12 transmembrane helices with both C- and N- termini located in the cytoplasm as previously described[42] (Fig. 3c, d, Supplementary Figs. 4e, 6a, b).

**Fig. 2 | Functional characterization of LysP. a** [³H]L-lysine counterflow uptake by reconstituted LysP. Blue shows transport into proteoliposomes containing LysP and red shows transport into liposomes with no LysP (control). Error bars represent the mean ± standard error of the mean (s.e.m.) of three independent experiments for proteoliposomes and two for liposomes. **b** IC₅₀ curves for the competitive inhibition of [³H]L-lysine uptake by external cold L-lysine (blue) and L-arginine (green) in LysP proteoliposomes preloaded with 1 mM of the respective amino acids. Uptake activity was normalized after subtraction of non-specific uptake, estimated from protein-free liposomes. Error bars represent the mean ± s.e.m. from three independent experiments. **c** Inhibition of [³H]L-lysine transport into proteoliposomes reconstituted with LysP. The inhibition was carried out in the presence of 10 mM of the different cold substrates. Error bars represent the mean ± s.e.m. of four independent experiments. **d** Binding of L-lysine and L-arginine to LysP by microscale thermophoresis (MST). Error bars represent the mean ± s.e.m. of three independent experiments. **e** Binding of thialysine to LysP by MST. Error bars represent the mean ± s.e.m. of three independent experiments. **f** pH gradient test: uptake of ³[H]L-lysine in reconstituted liposomes with LysP (proteoliposomes) and without LysP (control liposomes) at pH 4 inside and outside (red) and at pH 7 inside and pH 4 outside (blue). Error bars represent the mean ± s.e.m. of four independent experiments. **g** Sodium gradient test: uptake of ³[H]L-lysine in reconstituted liposomes with LysP (proteoliposome) with (blue) and without the addition of 50 mM NaCl (red). Error bars represent the mean ± s.e.m. of four independent experiments. Source data for Fig. 2 are provided in the Source Data file.

This cryo-EM structure displayed a 5 + 5 inverted repeat topology in which the first 5 transmembrane helices (TM1 - TM5) are a repeat of the next 5 (TM6 - TM10), with TM1 and TM6 broken in the middle[39,43] (Supplementary Fig. 6b). TM2 is linked to TM3 by a lateral helix (LH) located in the cytosol (Supplementary Fig. 6b). TM10 and TM11 are linked by a hydrogen-bonded turn followed by a bend located in the cytoplasm (Supplementary Fig. 6b). Unique to LysP, TM11 and TM12 are broken helices which have not been reported previously for transporters in the APC family (Supplementary Fig. 6b).

### Lysine binding site in LysP

The structure of the LysP-Nb5755 complex was determined in complex with L-lysine at 3.68 Å resolution by cryo-EM (Fig. 3c, d). In the complex, the L-lysine 'substrate' is contacted by residues on TM1, TM3, TM6 and TM10 (Fig. 4b). Specifically, L-lysine is coordinated by cation-π interactions with Trp105 and by hydrophobic interactions with Phe215 (Fig. 4c). In addition, and unlike the other amino acid transporters, the ε-amino group of L-lysine forms hydrogen bonds with polar uncharged amino acids, that include Asn104 on TM3 and Ser377 on TM10 (Fig. 4b). In this study, we observed that the α-carboxyl group of the L-lysine substrate forms hydrogen bonds with Glu112 on TM3 and with Gly33 on TM1, while the α-amino group is coordinated by Ser216 and Gln218 on TM6 (Fig. 4b). To validate the L-lysine binding site, site-directed mutagenesis was performed on key coordinating residues, primarily by substituting them with alanine. Size-exclusion chromatography of the purified mutant proteins showed that the mutations did not induce aggregation (Supplementary Fig. 4f). These mutations abolished L-lysine transport, confirming the functional relevance of those residues in the identified binding site (Fig. 4d, e). While density and mutagenesis analyses identified residues critical for lysine binding (Fig. 4), the role of an additional residue outside the binding pocket, such as Lys162, is considered separately below in relation to proton coupling (Fig. 4e).

Docking studies with thialysine into LysP revealed that it can occupy the same density as lysine in our cryo-EM structure (Supplementary Fig. 7a, b). Notably, the residues that coordinate lysine in our structure also interact with thialysine (Supplementary Fig. 7a, b). Thialysine is a close structural analogue of L-lysine, differing only by the substitution of the γ-methylene group with a sulfur atom. This alignment of binding interactions indicates that thialysine binding corroborates our experimental findings (Fig. 2c, e), providing additional evidence for its role as an effective competitive inhibitor for the transport of L-lysine by LysP.

### Role of Lys162 in proton coupling

We next examined Lys162, a residue located outside the lysine binding pocket. Substitution with alanine abolished L-lysine transport (Fig. 4e), suggesting a role in proton coupling rather than substrate recognition. Lys162 aligns with Lys158 in *Methanocaldococcus jannaschii* ApcT, implicated in proton translocation (Fig. 4f), and occupies the equivalent position of the second sodium-binding site in LeuT[23]. Together, these parallels lead us to hypothesize that Lys162 contributes to coupling proton flux to substrate transport, a role that will require further experimental validation.

### Molecular basis of L-lysine selectivity by LysP

Binding and transport studies of LysP showed that it specifically binds and transports L-lysine (Fig. 2c, d and Supplementary Fig. 2, 3). To understand the molecular underpinnings for the specificity of LysP towards L-lysine, sequence and structural alignments were performed with a close homologue of the mammalian Cationic Amino acid Transporter (CAT) transporter family from *Geobacillus kaustophilus* (GkApcT)[39] and the *Escherichia coli* arginine-agmantine antiporter from the APC transporter family (EcAdiC)[25,26] (Fig. 5a–c). Both GkApcT and EcAdiC transport L-arginine. The structures of each of these transporters were solved in complex with L-arginine. The residues that coordinate L-arginine in both (Supplementary Fig. 7c, d)[39] are different from those that coordinate L-lysine in LysP (Fig. 4b). Of note is the fact that Trp105 in LysP which is homologous to Tyr116 in GkApcT and to Cys97 in EcAdiC, both reside in TM3 (Fig. 5b, c). We propose that LysP uses Trp105 as a gate that sterically clashes with the guanidinium group of arginine during the transport cycle in LysP, thus essentially shutting down transport of L-arginine (Fig. 5b). By contrast, the ε-amino group of L-lysine is accommodated in the binding pocket because it is smaller than the guanidinium of arginine (Fig. 5b, c). Structural analysis of EcAdiC revealed that the guanidinium group of L-arginine forms a hydrogen bond with Cys95 on TM3, while Trp293 on TM6 acts as the gating residue in the occluded state[25]. (Fig. 5c and Supplementary Fig. 7c).

Additionally, the α-amino group of L-lysine forms a hydrogen bond with Ser377, located 3.0 Å away in LysP (Fig. 5d). Since Ser377 is not conserved among cationic amino acid transporters, we propose that it contributes to the substrate specificity of LysP for L-lysine.

Finally, the L-lysine in LysP is sandwiched between Phe215 and Trp105 (Fig. 4c). While Trp105 is proposed to act as a gating residue in the inward-occluded state, we hypothesize that Phe215 acts also as a gating residue, this time in the outward-open state to prevent the release of substrate back into the periplasm. In EcAdiC, Trp202 which is homologous to Phe215 in LysP, has been reported to be the gating residue in the outward-open state[25,26]. Both are located on TM6.

### LysP−L-Lysine−nanobody CA5755 binding interface

The LysP loops on the periplasmic surface of the transporter interact extensively with the Nb5755 nanobody used for structure determination, burying 793 Å² and 882 Å² of surface area on the nanobody and LysP, respectively. These values are typical for buried protein−protein interfaces (Fig. 6a, b). The nanobody contributes -11 residues from CDR1, CDR2, and CDR3, while LysP provides 29 residues from loop and helical regions at the interface. The interaction is stabilized by several hydrophobic patches formed by aliphatic and aromatic residues, which undoubtedly contribute to the strong binding affinity corroborated by the 20 °C increase in 'melting temperature' recorded upon complex formation (Supplementary Fig. 4a).

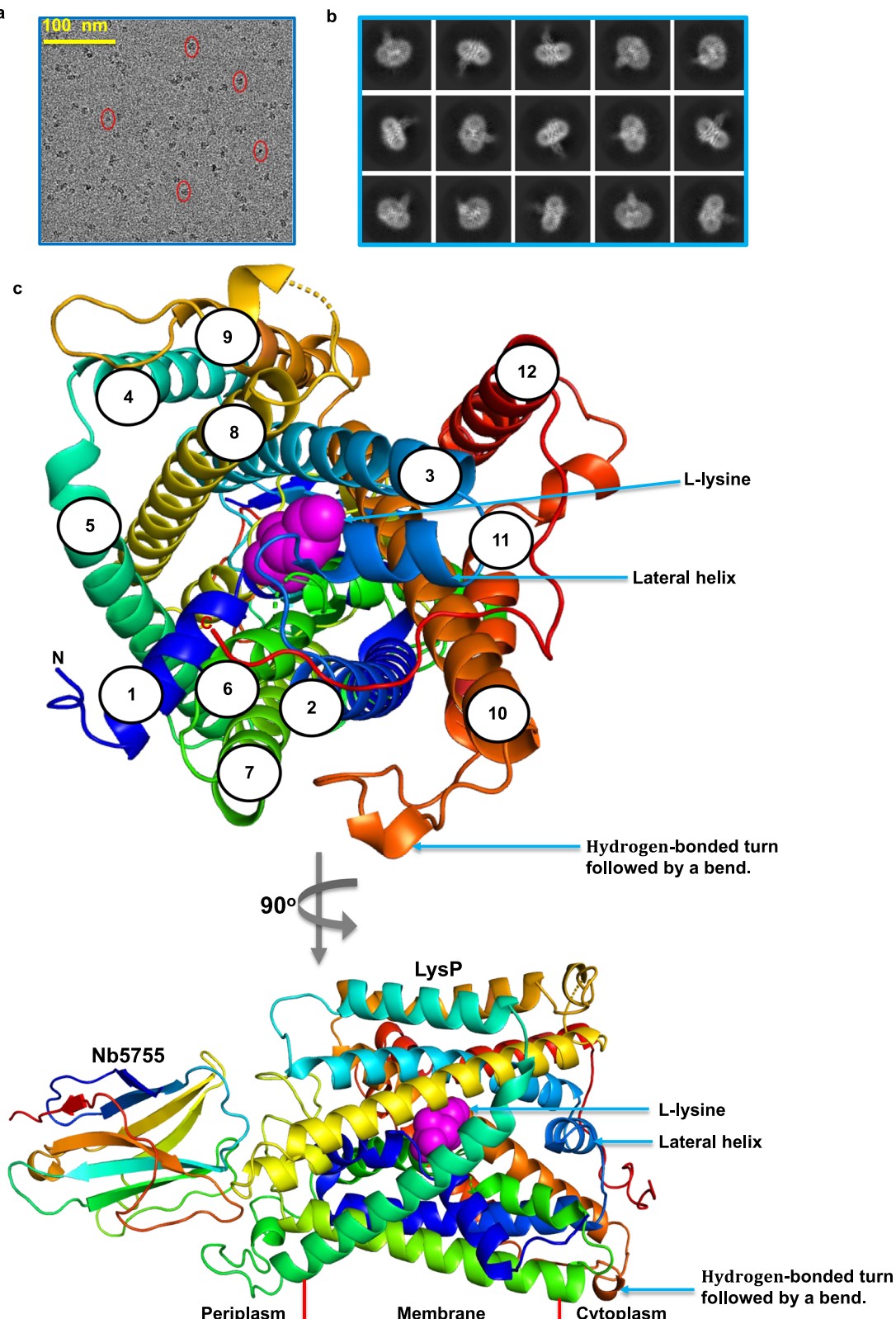

**Fig. 3 | Structure of LysP-Nb5755-L-lysine complex. a** Representative cryo-EM micrograph illustrating extraction boxes (red ovals) indicating the particle regions selected for further analysis. **b** Representative 2D classes of LysP-Nb5755-L-lysine. **c** Cytoplasmic view of the LysP-Nb5755 structure in complex with L-lysine. LysP has 12 transmembrane helices with both C- and N-termini located in the cytoplasm.

Helices 1 and 6 are broken in the middle, helices 2 and 3 are linked by a lateral cytoplasmic helix. Helices 10 and 11 are linked by a hydrogen-bonded turn followed by a bend. **d** View from the membrane of the LysP-Nb5755 structure in complex with L-lysine corresponding to a 90° rotation of the cytoplasmic view shown in **c**.

**Table 1 | Cryo-EM data collection, refinement and validation statistics**

| | LysP-L-lysine-Nanobody 5755 complex |
|---|---|
| **Data collection and processing** | |
| Magnification | 130000 |
| Voltage (kV) | 300 |
| Electron exposure (e–/Å$^2$) | 38.7 |
| Defocus range (μm) | −0.8 to −2.4 |
| Pixel size (Å) | 0.6725 |
| Symmetry imposed | None |
| Initial particle images (no.) | 177424 |
| Final particle images (no.) | 78459 |
| Map resolution (Å) | 3.68 |
| FSC threshold | 0.143 |
| Map resolution range (Å) | 3.2 to 5.3 |
| **Refinement** | |
| Initial model used (PDB code) | AlphaFold2 structure model |
| Model resolution (Å) | 3.7 |
| FSC threshold | 0.5 |
| Model resolution range (Å) | 3.2 to 4.1 |
| Map sharpening $B$ factor (Å$^2$) | −138 |
| Model composition | |
| Non-hydrogen atoms | 4568 |
| Protein residues | 595 |
| Ligands | — |
| $B$ factors (Å$^2$) | |
| Protein | 139.0 |
| Ligand | 124.0 |
| R.m.s. deviations | |
| Bond lengths (Å) | 0.0048 |
| Bond angles (°) | 0.68 |
| Validation | |
| MolProbity score | 1.86 |
| Clashscore | 5.14 |
| Poor rotamers (%) | 1.72 |
| Ramachandran plot | |
| Favored (%) | 93.89 |
| Allowed (%) | 6.11 |
| Disallowed (%) | 0 |

In addition to hydrophobic contacts, the interface is rich in polar interactions, that include six hydrogen bonds and three salt bridges. Notably, Arg26 from the nanobody forms one hydrogen bond and two salt bridges with Glu288 in LysP, while nanobody Arg52 engages Asp277 in LysP electrostatically and also forms hydrogen bonds with Asn269 and Asp277 in the transporter. Additional polar contacts involve Tyr31n, Asn76n, and Phe28n of the nanobody interacting with Asn19, Ser276, and Arg289 of LysP, respectively (Supplementary Fig. 4a).

## Discussion

The structural and functional analysis of LysP in *P. aeruginosa* has revealed three key molecular aspects of the transporter's function and pharmacological potential.

### LysP selectively transports L-lysine

The cryo-EM structure of the LysP–Nb5755 complex was obtained with bound L-lysine in an inward-occluded state (Fig. 3c, d, and

Supplementary Fig. 4e). Indeed, most amino acid transporters exhibit broad spectrum substrate specificity also referred to as substrate promiscuity. We propose that the L-lysine specificity of LysP derives primarily from hydrogen bonds between the substrate and the polar, uncharged side chains of Ser377 and Asn104 in LysP, as well as a cation–π interaction between the ε-amino group of L-lysine and the gating residue, Trp105, located on TM3 of the transporter (Fig. 5d, e). Like L-lysine, L-arginine is a cationic amino acid. However, it does not bind to nor is it transported by LysP. This disparity can be explained by the fact that the guanidinium group of arginine is bulkier than the ε-amino group of L-lysine. It likely introduces a steric clash with the gating, Trp105, which does not happen with L-lysine. This proposal is supported by the in vitro MST study in which L-arginine binding was more than a thousand-fold weaker than that of L-lysine (Supplementary Fig. 1a, c). It was also corroborated by the transport inhibition study where L-arginine did not inhibit the transport of L-lysine at concentrations up to 10 mM (Fig. 2c). We also propose that the residues coordinating the ε-amino acid side chain of L-lysine may play a crucial role in determining binding affinity and specificity. Indeed, in LysP, the ε-amino group is predominantly coordinated by polar uncharged amino acids (Fig. 5d). This coordination pattern contrasts with that observed in the L-arginine transporters, EcAdiC and GkApcT, as illustrated in Supplementary Fig. 7c, d. Finally, to support the specificity of LysP for L-lysine, transport assays revealed that no proteinogenic amino acid, with the exception of cold L-lysine, was able to compete for the transport of tritiated L-lysine (Fig. 2c).

### Proposed lysine-specific uptake mechanism across bacterial cell membrane for extreme pH regulation and survival

The specific transport of L-lysine by LysP is likely a survival mechanism. Exposure to extremely low pH activates L-lysine transport in bacteria. L-Lysine subsequently upregulates the *cadAB* operon leading to lysine decarboxylation in a reaction that consumes a proton. The resulting cadaverine is exported from the cytoplasm in exchange for extracellular lysine via the antiporter, maintaining continuous lysine uptake. This exchange consumes intracellular protons and contributes to the neutralization of external acidity, thereby promoting bacterial survival under acidic stress (Fig. 1a). Indeed, induction of the *cadAB* operon requires the presence of L-lysine, which is transported into the cytoplasm by LysP[13]. Additionally, the transcriptional activator CadC, upon sensing low periplasmic pH, interacts with LysP to trigger operon expression[14].

An analysis of LysP structure suggests that this lysine transporter shares a similar alternating access transport mechanism with GkApcT[39] and EcAdiC[24–26]. The structure of LysP in the inward-occluded conformation was analyzed with the help of normal mode analysis (NMA) to produce plausible conformations of LysP in the outward-open, outward-occluded, occluded, and inward-open conformations of LysP. This analysis suggests that, in the outward-open state, the extracellular tunnel of LysP is open and accessible to a proton and to an L-lysine molecule and that it channels them to the binding site within the transporter (Fig. 7a, b, and Supplementary Movie 1). The binding of L-lysine and the protonation of the ε-amino group of Lys162 in TM5 results in the movement of TM6b and TM1a towards the binding site enabling Phe215 on TM6 to reposition so as to prevent the release of substrate back into the periplasm (Fig. 6b and Supplementary Movie 1). This conformational change facilitates closure of the extracellular gate and the formation of the occluded state (Fig. 6b and Supplementary Movie 1), followed by a cation-π interaction between the ε-amino group of the L-lysine substrate and Trp105 on TM3 and strong hydrogen bond formation with Ser377 and Asp104 (Fig. 5d, e). These varied interactions trigger further conformational changes that drive LysP into a fully occluded state and that lead to an opening of the intracellular gate. Gate opening to the cytoplasm is facilitated by proton release to the cell's interior from Lys162. This, in turn, causes

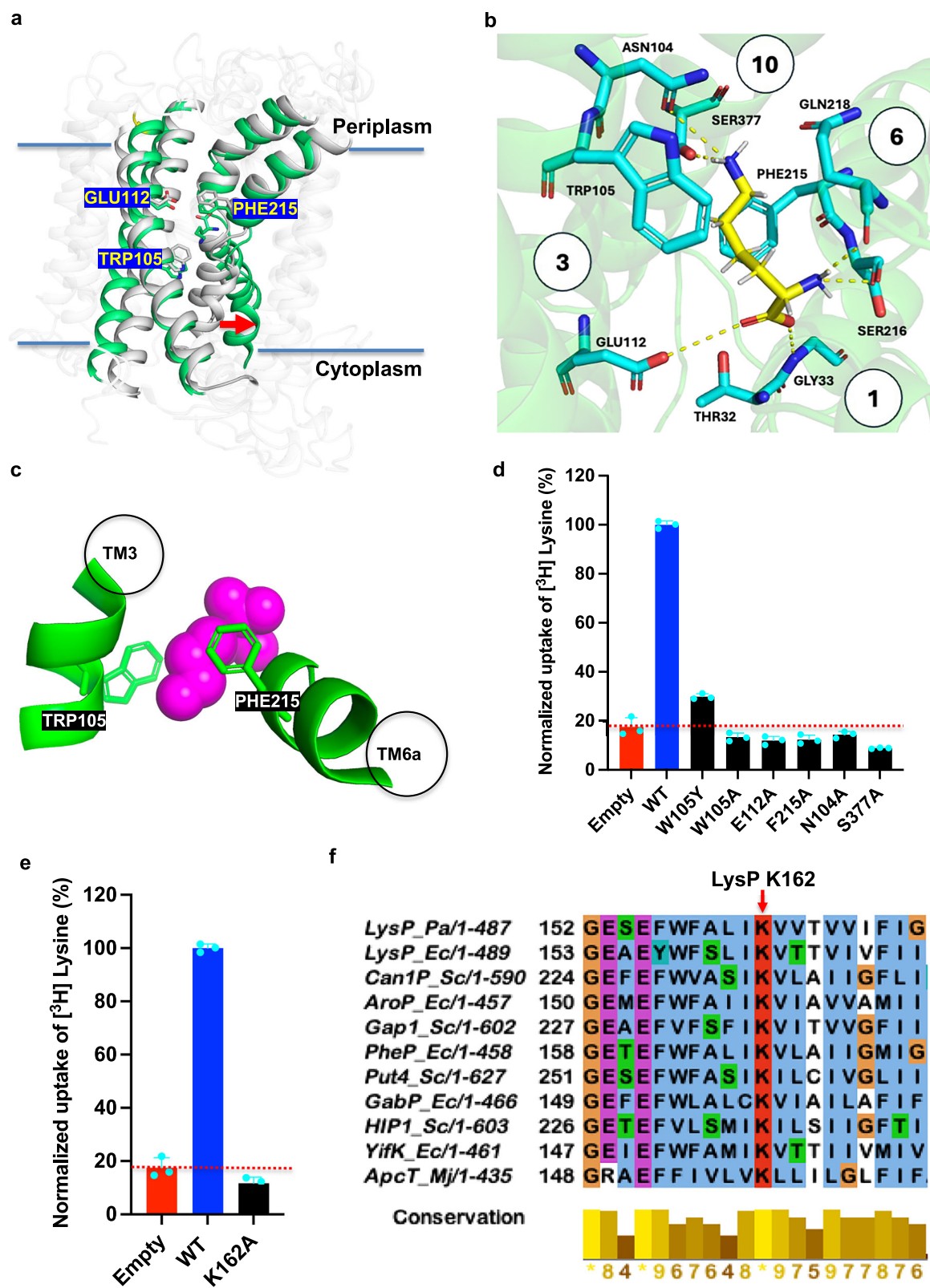

movement of TM1a and TM6b away from the cytoplasmic tunnel setting the system up for release of L-lysine and the proton into the cytoplasm (Fig. 6a, b and Supplementary Movie 1). Our cryo-EM structure captures the inward-occluded state in which the cation–π interaction between Trp105 on TM3 of LysP and the ε-amino group of the L-lysine substrate, as well as hydrogen bonds involving Ser377 and Asn104, remain intact (Fig. 6a, b). Finally, in our inward-occluded state

structure, TM1a and TM6b have moved away from the cytoplasmic tunnel to open the binding site in preparation for the release of L-lysine and a proton to the cytoplasm (Fig. 6a, b). Structured water has been identified as crucial for stabilizing EcAdiC and for shaping the substrate-binding site with water molecules acting as placeholders for substrate atoms[44]. Likewise, it is possible that water binding plays a role in LysP transport. However, at a resolution of 3.7 Å, structured

**Fig. 4 | L-Lysine binding site of *Pseudomonas aeruginosa* LysP.**
**a** Superimposition of the AlphaFold2-predicted outward-open structure (grey) and the inward-occluded cryo-EM structure of LysP (green). Red arrow indicate the movement of TM1a and TM6b, suggesting conformational changes that facilitate L-lysine release into the cytoplasm. **b** L-Lysine binding site of LysP viewed from the cytoplasm. The L-lysine is coordinated to LysP by residues on TM1, TM3, TM6 and TM10 through hydrophobic interactions, a cation π-interaction, and by hydrogen bonding (yellow dashed lines). **c** Membrane view of the gating residues showing L-lysine sandwiched between Trp105 on TM3 and Phe215 on TM6. **d** Effect of mutations in the binding site residues (shown in **b**) on L-lysine transport. Error bars represent the mean ± s.e.m. of three independent experiments. **e** Effect of mutation of the proposed proton-coupling residue K162 on L-lysine transport. Error bars

represent the mean ± s.e.m. of three independent experiments. **f** Sequence alignment of cationic amino acid transporters (*Pseudomonas aeruginosa* LysP, UniProt: Q9HVG3, *Escherichia coli* LysP, UniProt: P25737, *Saccharomyces cerevisiae* Can1P, UniProt: C7GXF0, *Escherichia coli* AroP, UniProt: P15993, *Saccharomyces cerevisiae* Gap1, UniProt: P19145, *Escherichia coli* PheP, UniProt: P24207, *Saccharomyces cerevisiae* Put4, UniProt: P15380, *Escherichia coli* GabP, UniProt: P25527, *Saccharomyces cerevisiae* HIP1, UniProt: P06775, *Escherichia coli* YifK, UniProt: P27837, and *Methanocaldococcus jannaschii* ApcT, UniProt: Q58026), highlighting the conserved lysine residue in *P. aeruginosa* LysP (K162) involved in proton coupling, corresponding to the second sodium-binding site (Na2) in LeuT. Source data for Fig. 4 are provided in the Source Data file.

waters cannot be reliably identified in our current cryo-EM structure of LysP.

## Antibiotic Target Potential

Previous studies have shown that supplementation with specific amino acids such as lysine, arginine, or glutamate can rescue the growth of bacteria exposed to extremely low pH conditions[43]. LysP mediates lysine uptake and, by interacting with the membrane-bound sensor CadC, contributes to the activation of *cadBA* operon expression[10-14]. This leads to the expression of CadA, a lysine decarboxylase that consumes protons during the conversion of lysine to cadaverine. Simultaneously, CadB is upregulated, promoting cadaverine export in exchange for extracellular lysine[10-14]. Critically, the expression and function of both CadA and CadB are dependent on the availability of L-lysine and the interaction between LysP and CadC[14]. Disruption of lysine transport by LysP and/or its regulatory interaction with CadC could therefore compromise a key acid resistance mechanism (Fig. 1a).

Building on the structural and functional insights provided in this study, we propose that LysP represents a potential antibiotic target in *Pseudomonas aeruginosa* and possibly other bacterial species (Fig. 1b, c). Inhibiting LysP could impair bacterial adaptation to acidic environments by preventing lysine-dependent activation of acid neutralization pathways.

In this study, we identified L-4-thialysine, a lysine analog known to interfere with bacterial translation[45-47] as a competitive inhibitor of lysine transport by LysP (Fig. 1c), with a $K_i$ of ~31 μM. Thialysine is a close L-lysine analogue and, not surprisingly, is transported by LysP. Once inside the cell, thialysine can be incorporated into proteins in place of lysine[48-50], leading to protein instability and degradation[51]. Although thialysine primarily targets lysyl-tRNA synthetase[46] and not LysP directly, its ability to competitively inhibit lysine uptake highlights LysP's vulnerability to small-molecule interference. Notably, thialysine has been shown to completely inhibit *E. coli* growth at concentrations as low as 5 μM[46].

While the findings of this study are promising with regard to antibiotic development, we acknowledge that LysP has not yet been experimentally validated as an essential or druggable target. Further studies are needed to assess whether inhibition of LysP alone is sufficient to suppress bacterial growth, particularly under lysine-limited or acidic conditions. Nonetheless, our results provide a framework for exploring LysP as a novel antimicrobial target and sets the stage for further investigation into lysine transport and acid resistance pathways as potential points of therapeutic intervention. If LysP is not shown to be essential, it may well turn out to be a virulence factor. Interest in targeting virulence factors as opposed to essential proteins has grown of late given the reduced likeliness of resistance development when virulence factors are the targets.

## Methods
### Cloning and overexpression of LysP
The gene encoding the lysine specific permease (*lysP*) from *P. aeruginosa* strain PAO1 was cloned into pET200/D-TOPO (Invitrogen),

digested with NdeI and BamHI restriction enzymes and then subcloned into the standard pET 28(a) derived GFP-His$_8$ fusion vector (pWaldo-GFPd)[52] between NdeI and BamHI restriction sites. The forward and reverse primers used for the PCR amplification were 5′-CACCCATATGACTGACCTGAACACCAGCCAG-3′ and 5′-GGATCCGG-TATTGGTCGGGCTGACGTC-3′, respectively. The template DNA source used for the PCR reaction was from *P. aeruginosa* strain PAO1 cells grown in Luria Bertani (LB) broth medium (Fluka). The inclusion of the *lysP* gene in the pWaldo-GFPd vector was validated by DNA sequencing, using T7 forward and reverse primers (Eurofins MWG GmbH, Anzinger Strasse 7a, D-85560 Ebersberg, Germany).

A single colony of freshly transformed *Escherichia coli* (DE3) C43 cells with pWaldo-GFPd vector harboring the *lysP* gene from *P. aeruginosa* strain PAO1 grown on an LB agar/kanamycin plate was used to inoculate 50 mL of LB supplemented with 50 μg/mL of kanamycin and grown for 16 h at 37 °C with shaking at 200 r.p.m using the INFORS HT multitron standard shaker. The culture was then used to inoculate (1/50 dilution) 3 ×1 L of Terrific Broth media supplemented with 50 μg/mL of kanamycin and grown at 37 °C with shaking at 200 r.p.m using the INFORS HT multitron standard shaker. The temperature of the shaker was dropped from 37 °C to 25 °C after induction with 0.4 mM isopropylthio-β-galactoside (IPTG) at an OD$_{600}$ of 0.6. The cells were then grown at 25 °C with shaking at 200 r.p.m for a further 22 h. GFP fluorescence (excitation 485 nm and emission 512 nm) was read on 100 μL samples taken 22 h after IPTG induction using the SpectraMax Gemini EM plate reader (Molecular Devices).

### Membrane isolation and purification of LysP
3 L of over-expressed cells were harvested immediately by centrifugation (Beckman Coulter Avanti J-20 XPI centrifuge, Beckman JLA-8.100 rotor) at 6000 × g for 10 min at 4 °C. Supernatants were discarded and cell pellets were resuspended in ice cold 1 × phosphate buffered saline at pH 7.4 (PBS). Prior to cell lysis, 1 mM MgCl$_2$, pefabloc (1 mg/mL final concentration) and DNase (20–100 U/mL final concentration) was added to the cells and the cells were broken using a Constant System cell disruptor (2 passes at 30 kPSI). The unbroken cells and debris were removed by centrifugation at 20,000 × g for 10 min at 4 °C (Beckman Coulter Avanti J20 XPI centrifuge, JLA-16,250 rotor). The supernatant containing the membrane fraction was transferred to Beckman ultracentrifuge tubes and subjected to ultracentrifugation for 2 h at 41,000 r.p.m (Beckman Optima L-100 XP ultracentrifuge, Beckman 45 Ti rotor). Membrane pellets were resuspended to a total protein concentration of 3.5 mg/mL in ice cold 1 × PBS using the Bicinchoninic Acid (BCA) protein assay kit (Pierce), rapidly frozen in liquid nitrogen and stored at −80 °C until used.

Membrane suspensions were thawed at room temperature (22 °C) and then diluted to a total protein concentration of 3.0 mg/mL in solubilization buffer (1 × PBS, 10%(v/v) glycerol, 150 mM NaCl and 1% (w/v) of n-dodecyl-β-D-maltopyranoside (DDM)) and further incubated for 1 h at 4 °C with gentle stirring. Non-solubilized materials were removed by centrifuging the sample at 41,000 r.p.m for 1 h at 4 °C using rotor type

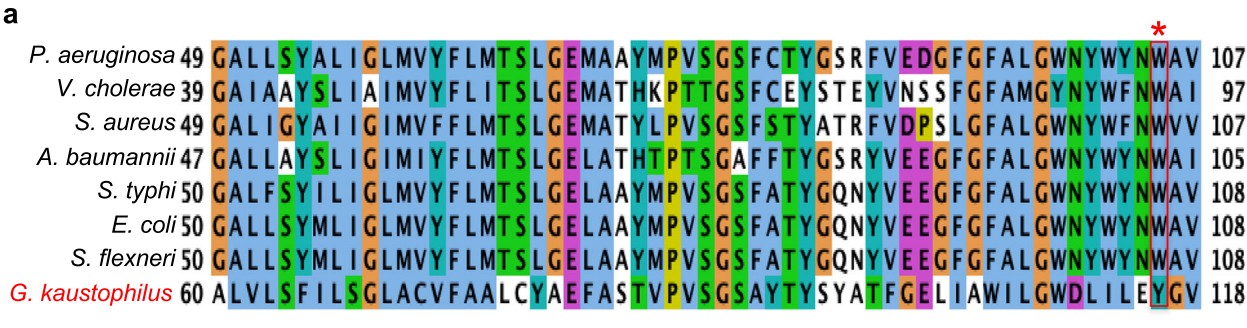

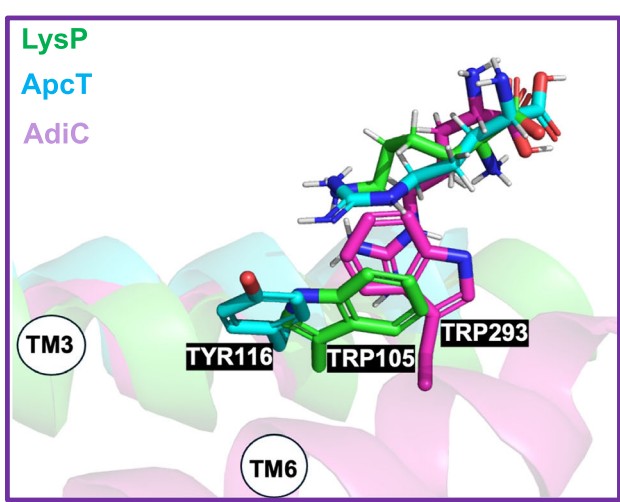

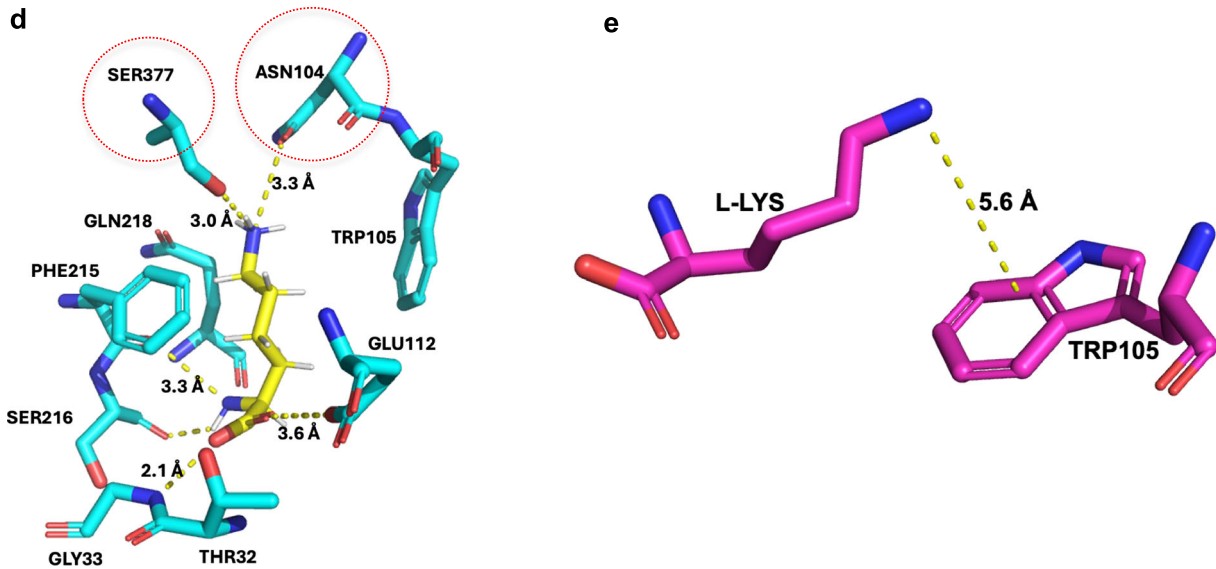

**Fig. 5 | Molecular basis of L-lysine selectivity by LysP. a** Sequence alignment of LysP from different bacterial species and *Geobacillus kaustophilus* ApcT. W105 in LysP is homologous to Y116 in ApcT. **b** Structural comparison of LysP (green), ApcT (cyan) (PDB: 6f34) and AdiC (pink) (PDB: 3L1L) showing the cation-π interaction between L-lysine in LysP and Trp105 on TM3. LysP Trp105 is homologous to ApcT Tyr116 on TM3 which forms a cation-π interaction with L-arginine. In AdiC, the corresponding residue is Cys95 on TM3 which forms a hydrogen bond with the guanidinium group of L-arginine instead. **c** Structural comparison of LysP (green), ApcT (cyan) and AdiC (pink) highlighting the corresponding gating residue, Trp293 on AdiC located on TM6 instead of TM3 as in LysP. **d** Proposed residues involved in lysine specificity—Ser377 and Asn104—form strong hydrogen bonds with the side-chain amino group of the substrate, L-lysine. **e** Cation-π interaction between the ε-amino group of L-lysine and the gating residue, Trp105 in LysP.

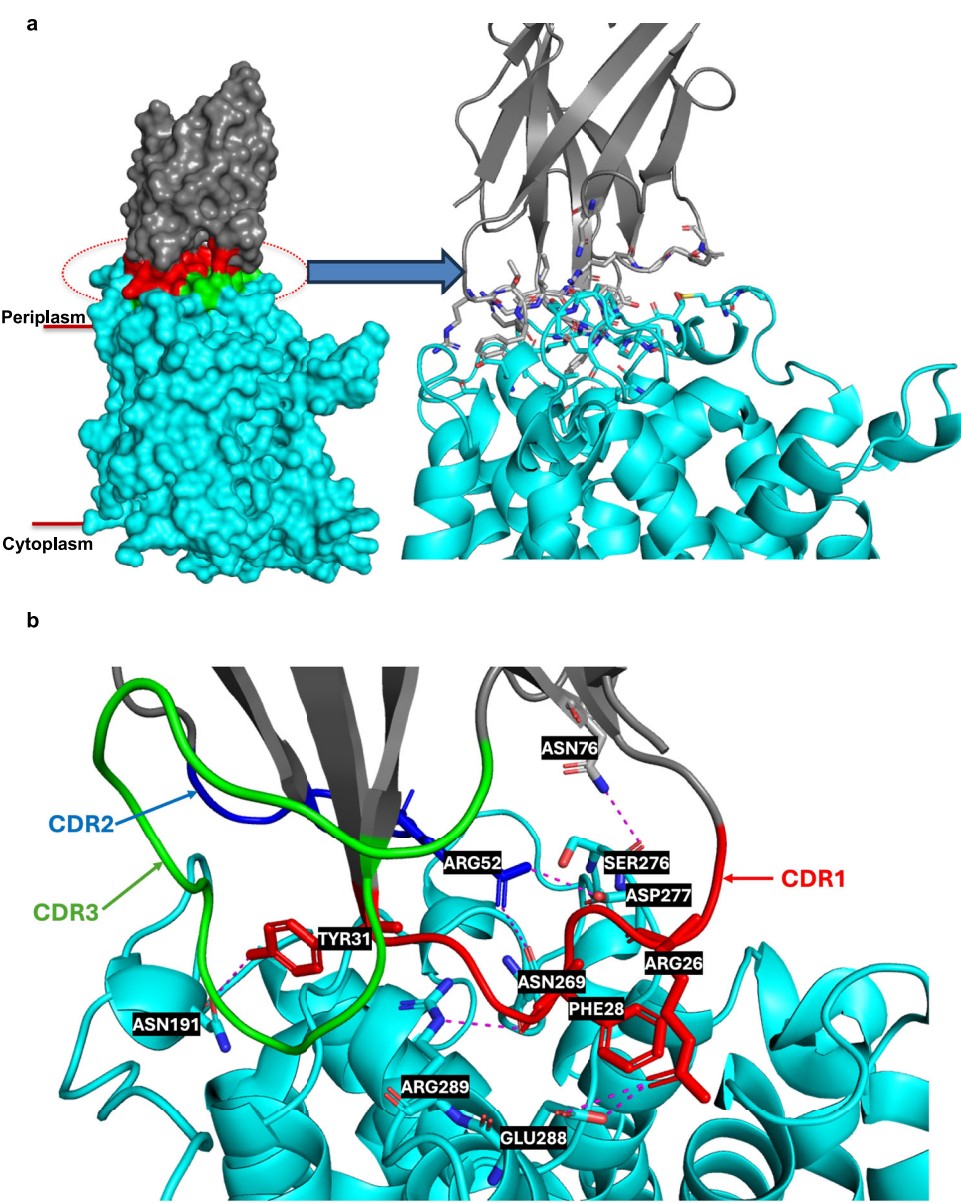

**Fig. 6 | The LysP-L-Lysine-Nb5755 interaction. a** Membrane view showing the LysP-Nb5755 binding interface. **b** Expaned view of the LysP-Nb5755 binding interface showing residues in LysP interacting with residues in the CDR1, CDR2 and CDR3 of Nb5755.

45 Ti in the OptimaTM L-100 XP Ultracentrifuge (BECKMAN COULTER). The resulting supernatant containing the solubilized LysP-GFP was incubated with Ni²⁺-NTA resin (Qiagen) for 2 h at 4 °C using 1 mL of resin per mg GFP. To reduce non-specific binding, 10 mM imidazole pH 7.5 was added to the sample. The slurry was loaded on to a glass Econo gravity column (Bio-Rad) and washed with 20 column volumes of solubilization buffer containing 10 mM imidazole pH 7.5. The column was washed with a further 20 column volumes each of solubilization buffer containing 20 and 35 mM imidazole pH 7.5 at 4 °C. The protein was then eluted from the column and the eluate collected using 40 mL elution buffer (solubilization buffer containing 250 mM imidazole pH 7.5) at 4 °C. The eluate was incubated overnight at 4 °C with an equal amount of His-tagged Tobacco etch virus (TEV) protease (1 mg TEV protease/mg GFP) was added into a dialysis tube (14 kDa MWCO) and the sample was dialysed against 3 L of dialysis buffer containing 20 mM Tris HCl pH 7.5, 150 mM NaCl and 0.003% (w/v) lauryl maltose neopentyl glycol (LMNG) at 4 °C with gentle stirring of the dialysis buffer with a magnetic rod/stirrer. The

sample consisting of GFP-free LysP, GFP, TEV and likely a small amount of uncleaved LysP-GFP was filtered using 0.22 μm Millipore filters to remove protein aggregates and was further subjected to the second round of IMAC to remove the free GFP, TEV protease and any uncleaved LysP-GFP. In this reverse IMAC step, a 5 mL Ni²⁺-NTA HisTrap fast flow column (Qiagen) was equilibrated with 10 mL of buffer containing 20 mM Tris HCl pH 7.5, 150 mM NaCl and 0.003% (w/v) LMNG at a flow rate of 0.2 mL/min at 4 °C and the eluent containing GFP-free LysP was collected and concentrated at 3500 × g at 4 °C using rotor C0650 in the AllegraTM X-22R centrifuge (BECKMANN COULTER Clare, Ireland) for 5 min at 4 °C using a Centricon 50 kDa molecular weight cut off (MWCO) concentrator (Millipore: MA, USA, Lot R0KA67379). After the 5 min spin, the sample was resuspended by pipetting up and down. The concentrated sample (0.5 mL final volume) was loaded onto a Superdex 200 increase 10/300 size exclusion column (GE Healthcare) at a flow rate of 0.4 mL/min and the absorbance at 280 nm was monitored. Prior to loading the sample, the column was pre-equilibrated with buffer

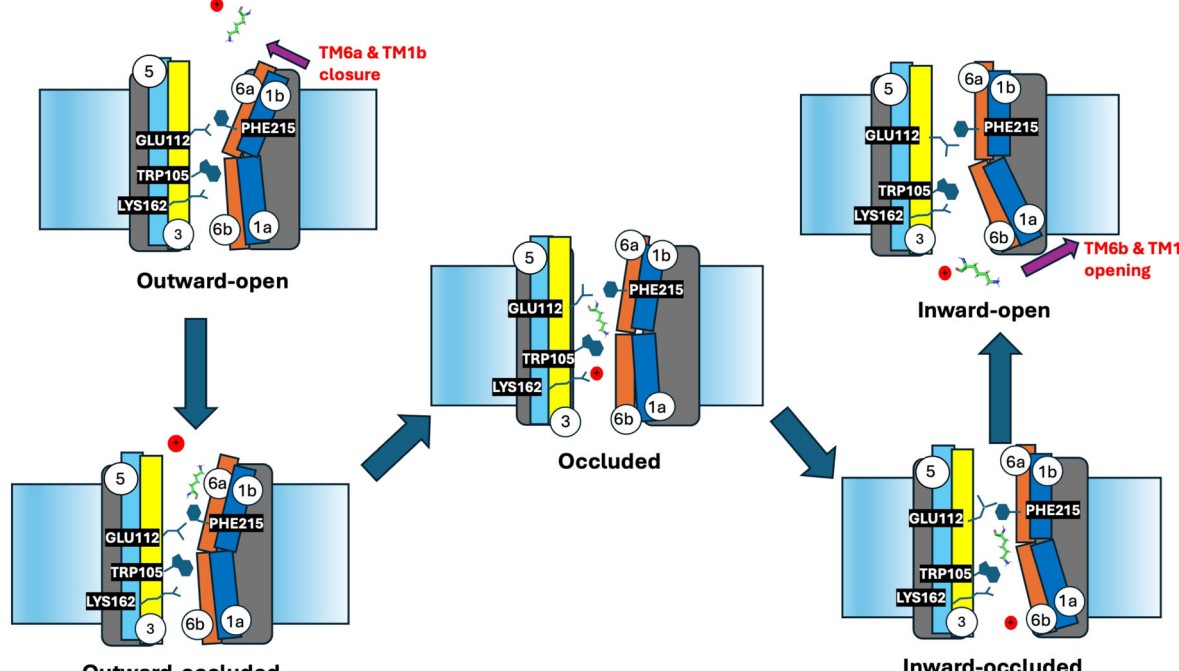

containing 10 mM Tris HCl pH 7.5, 100 mM NaCl and 0.003% (w/v) LMNG. For transport assays, the 0.003%(w/v) LMNG in the gel filtration buffer was substituted with 0.03%(w/v) DDM. LysP was concentrated using 100 kDa molecular weight cut off concentrator (Millipore). The concentration of purified GFP-free LysP was determined from the absorbance at 280 nm, using an extinction coefficient of 97,290 M$^{-1}$.cm$^{-1}$ calculated from the amino acid sequence.

## Circular dichroism

CD spectra of purified LysP at 0.6 mg/mL in 150 mM NaCl, 20 mM Tris pH 7.5 and 0.003% (w/v) LMNG, was recorded using a Jasco J-815 CD Spectrophotometer from 190 nm to 260 nm at a scan speed of 100 nm/min, a band width of 1 nm, a path length of 1 mm and an ellipticity of 0.1 degrees at 20 °C. The same parameters were used for the blank containing 150 mM NaCl, 20 mM Tris pH 7.5 and 0.003%

**Fig. 7 | Proposed mechanism of L-lysine transport by LysP. a** Surface view from the membrane highlighting key gating residues—Trp105, Phe215, and the proton-coupling residue Lys162—along with transmembrane helices TM1a, TM1b, TM3, TM5, TM6a, TM6b, and TM10, which together form the lysine transport pathway and constitute major gating elements. The L-lysine substrate is shown with cyan carbon atoms. **b** In the outward-facing conformation, the extracellular tunnel is open, allowing entry of a proton and L-lysine into the binding site. Binding of L-lysine and protonation of Lys162 on TM5 trigger conformational changes, including inward movement of TM6b and TM1a. This movement enables Phe215 on TM6 to contribute to sealing the binding pocket, leading to closure of the

extracellular gate and formation of an inward-occluded state. In this state, the substrate is trapped within the central binding site and stabilized by a cation–π interaction between the ε-amino group of L-lysine and Trp105 on TM3, along with hydrogen bonds to Ser377 and Asn104. These stabilizing interactions promote further rearrangements, particularly the outward movement of TM1 and TM6, resulting in opening of the intracellular gate. This transition defines the inward-open state, enabling L-lysine and the coupled proton to be released into the cytoplasm. The release is facilitated by deprotonation of Lys162, which resets the transporter for another cycle of substrate uptake.

---

(w/v) LMNG without LysP. Data were processed using the K2d method in the online CD analysis webpage (DIOCROWEB) (https://dichroweb. cryst.bbk.ac.uk/html/home.shtml)[53–55].

### Microscale thermophoresis (MST)

MST was carried out as described previously[56,57]. This experiment was carried out at NanoTemper Technologies GmbH, Floßergasse 4, 81369 Munich, Germany. In brief, serial 2-fold dilutions of each ligand were prepared starting at 800 μM for L-lysine, 6.4 mM for L-4-thialysine, 50 mM D-lysine, 200 mM of L-arginine and 100 mM each of L-histidine, glycine, L-proline, L-glutamine, L-glutamate, L-methionine, L-isoleucine, L-leucine, L-threonine, L-valine and L-alanine to obtain 16 different concentrations of ligand at 22 °C. All concentrations of ligands were mixed with protein in a 1:1 by volume ratio and 4 μL each were loaded into glass capillaries. The final protein concentration was 0.6 μM. MST analysis was carried out using Monolith Label Free instrument (Nano-Temper) using a laser power and LED sensitivity of 20% at 22 °C. All ligands and protein solutions used for MST were prepared in buffer containing 150 mM NaCl, 20 mM Tris pH 7.5 and 0.003% (w/v) LMNG.

### Reconstitution and counterflow transport assays

The reconstitution and counterflow assay begin with a lipid purification step. This involved dissolving 100 mg of *E. coli* polar lipid (Avanti) (white solid powder) in 2 mL of chloroform which was then mixed with 10 mL of acetone containing 2 μL of β-mercaptoethanol (14.7 M stock) under argon and the mixture was stirred for 16 h at 4 °C. The precipitated lipid was sedimented by centrifuging at 10,000 × g for 12 min at room temperature (22 °C) and the supernatant discarded. The lipid pellet was redissolved in 10 mL of diethyl-ether under argon containing 2 μL of β-mercaptoethanol (14.7 M stock) and centrifuged at 8000 × g for 10 min at room temperature. The supernatant containing the desired lipid was dried to a thin film by evaporating the diethyl ether using argon gas. The purified lipid was redissolved in chloroform to a final concentration of 10 mg/mL and stored under argon at −20 °C. Glass tubes were used throughout at room temperature as chloroform dissolves plastic tubes. The purified lipid extract stored in chloroform was dried under argon, and the resulting lipid film was rehydrated to a final concentration of 10 mg/mL in Reconstitution Buffer (50 mM potassium phosphate, pH 7.6; 20 mM cold L-lysine; 1 mM DTT) by repeatedly vortexing (30 s) and heating to 50 °C for 10 cycles, until the lipid film was completely dispersed. To aid rehydration a series of incubations at 50 °C and mixing were performed until the lipid film was completely dispersed. Unilamellar vesicles of size, 0.1 μm were prepared using an extruder (Lipex Biomembranes, Inc. Vancouver, British Columbia) at a pressure of 300 psi to pass the lipid suspension once through 0.4 μm polycarbonate Nucleopore™ Track Etch double membrane filters. It was then passed once through 0.2 μm polycarbonate Nucleopore™ Track Etch double membrane filters and 7 times through 0.1 μm pore size polycarbonate Nucleopore™ Track Etch single membrane filters. The extrusion was carried out with the extruder kept at 50 °C in a water bath. This produced a homogenous suspension of 0.1 μm diameter vesicles. The liposome samples were used immediately for reconstitution of LysP. For reconstitution,

β-octyl-glucoside (OG) was used to destabilise the pre-formed liposomes and to allow the insertion of LysP into the liposome membrane. 1 mL of liposome suspension (10 mg lipid/mL) in Reconstitution Buffer was mixed with β-OG to a final concentration of 1.25% (w/v) from a 10% (w/v) stock solution, and incubated at 4 °C for 15 min. Purified LysP (10 mg/mL in 150 mM NaCl, 20 mM Tris-HCl pH 7.5, and 0.03% (w/v) DDM) was then added at a lipid-to-protein weight ratio of 100:1, and the mixture was transferred to a 5 mL glass test tube. For the control (liposomes alone), an equal volume of Buffer A (10 μL) was added in place of purified LysP (10 mg/mL in 150 mM NaCl, 20 mM Tris-HCl pH 7.5 and 0.03% (w/v) DDM). Excess β-OG was removed by diluting the protein–liposome mixture with 70 mL of Reconstitution Buffer, followed by centrifugation at 100,000 × g for 1 h at 4 °C to pellet the proteoliposomes and liposomes. The supernatant was discarded carefully, the pellets, containing proteoliposomes or liposomes were resuspended in 15 μL Reconstitution Buffer on ice and used directly.

For counterflow assay, proteoliposomes or liposomes were diluted 1 in 25 (by vol) using Counterflow Buffer (CB, 50 mM potassium phosphate pH 7.6 and 1 mM DTT) and the counterflow assay[58] was initiated by adding 40 μL of [³H] L-lysine (stock, 50 μM, 100 μCi/mL). After the addition of [³H] L-lysine, samples (80 μL) were taken at different time points and applied to a 0.22 μm pore size MF-Millipore™ Membrane Filter. The filters were immediately washed under vacuum with 4 mL of ice-cold CB. The washed filter was transferred to a 20 mL scintillation vial and submerged in 10 mL emulsifier safe liquid scintillant. The level of radioactivity associated with the liposomes or proteoliposomes was measured by liquid scintillation counting. All protein batches used for reconstitution were purified in DDM. For Na⁺ gradient tests, 50 mM NaCl was added to the external CB with the internal CB kept constant (50 mM potassium phosphate pH 7.6 and 1 mM DDT). For H⁺ gradient tests, the external pH was kept constant at pH 4 using 25 mM citrate-phosphate buffer while the internal pH values employed were pH 4 and pH 7.0 using 25 mM citrate-phosphate buffer instead of 50 mM KPi pH 7.6. An inhibition study was carried out at external pH 4.0 and internal pH 7.0 in the presence of 10 mM external cold inhibitors (the twenty L-amino acids, L-4-thialysine and D-lysine) and samples were collected and analysed after 3 and 20 min.

For the radioactivity assay of binding pocket mutants, purified LysP-GFP and each corresponding GFP-fusion mutant were incorporated into liposomes using a freeze–thaw method followed by extrusion. A lipid mixture consisting of bovine brain extract 7 (Sigma-Aldrich) and cholesteryl hemisuccinate (Sigma-Aldrich) was prepared at final concentrations of 30 mg/mL and 6 mg/mL, respectively, in a buffer containing 10 mM Tris-HCl (pH 7.5) and 2 mM MgSO₄ (A.B). This suspension was rapidly frozen and subsequently thawed at room temperature, followed by sonication. The sample was then centrifuged at 16,000 g for 15 min to isolate the supernatant, which contained small unilamellar vesicles. Subsequently, 10–20 μL of protein (40 μg total) was added to 500 μL of liposomes. The mixture was subjected to extrusion using a 400-nm pore membrane (LiposoFast; AVESTIN), yielding large unilamellar proteoliposomes. These were pelleted by ultracentrifugation at 250,000 g for 30 min at 4 °C and resuspended in A.B to a final concentration of approximately 120 mg/mL. Control

liposomes lacking protein were prepared identically, substituting the protein addition with an equal volume of buffer.

For uptake measurements, 5 μL of proteoliposomes were diluted into 45 μL of A.B buffer (pH 6.5) containing [$^3$H]L-lysine (0.17 mM; American Radiolabeled Chemicals), and the mixture was incubated at 25 °C. After 30 s, transport was halted by adding 1 mL of ice-cold A.B, followed by rapid filtration through a 0.22-μm hydrophilic mixed cellulose filter (Millipore). Filters containing liposomes were washed with 6 mL of A.B, transferred to scintillation vials, and emulsified in 5 mL of Ultima Gold scintillation fluid (PerkinElmer) prior to counting using a TRI-CARB 4810TR scintillation counter (110 V; PerkinElmer).

For IC$_{50}$ determination, the same procedure was followed, with the radiolabeled substrate concentration held constant while varying the concentration of non-radioactive lysine. Disintegrations per minute (DPM) values from protein-free liposomes (measured at 2 min) were used as background and subtracted from each corresponding condition. Data were then normalized internally. The IC$_{50}$ values were obtained by fitting a nonlinear regression of [inhibitor] versus the normalized response with a variable slope using GraphPad Prism v.10.4.1.

### Electron microscopy sample preparation and data acquisition

The cryo-EM grids were prepared by applying 3 μL of the LysP-Nb5755 complex protein at 2 mg/mL in buffer containing 100 mM NaCl, 20 mM Tris-HCl pH 7.5, 0.03% (w/v) DDM and 10 mM L-lysine, to a glow-discharged Quantifoil R1.2/1.3 200/300-mesh copper/gold holey carbon grid (QUANTIFOIL, Micro Tools GmbH, Germany) and blotted for 3.0 s under conditions of 100% relative humidity and 4 °C before being plunged into liquid ethane using a Mark IV Vitrobot (FEI). Grids were screened in a 200 keV Glacios transmission electron microscope (Thermo Fisher) at Uppsala University, Sweden and the Membrane Protein Lab, Diamond Light Source Ltd., UK cryo-EM imaging facilities. Micrographs were acquired on a Titan Krios microscope (FEI) operated at 300 kV with a K2 Summit direct electron detector (Gatan) at the cryo-EM Swedish National Facility in Stockholm. SerialEM software was used for automated data collection following standard procedures[59]. A calibrated magnification of ×130,000 was used for imaging, yielding a pixel size of 0.67 Å on images. The defocus range was set from −0.8 to −2.4 μm. Each micrograph was dose-fractioned to 50 frames at a dose rate of 7–8 electrons.pixel$^{-1}$.s$^{-1}$, with a total exposure time of 8 s, resulting in a total dose of ~80 electrons.Å$^{-2}$.

### Cryo-EM data processing, model building, and structure analysis

The data set was processed using CryoSPARC[60]. The dose fractioned movie frames were aligned using "patch motion correction", the contrast transfer function (CTF) was estimated using "Patch CTF estimation", and particles were picked using automated blob picker and extracted by binning 6 times in cryoSPARC live. Subsequently, multiple rounds of 2D classification were performed. Finally, high-quality particles were selected and extracted with a binning size of 1.0 Å per pixel. A total of 177,424 particles were used for ab initio map generation, followed by heterogeneous refinement. After refinement, 145,063 particles were selected. Several rounds of non-uniform refinement and local refinement were performed. To remove bad particles, further phase randomized heterogeneous refinement was performed with low-pass filters at 8 Å, 20 Å and 40 Å. The final 78,459 particles were selected and local refinement with a tight mask was performed. The overall resolution reached 3.68 Å at the gold-standard Fourier shell correlation (FSC) resolution value of 0.143. The local resolution was calculated in CryoSPARC[60].

An AlphaFold 2[61] model of LysP was automatically fitted into the cryo-EM density map of our inward-facing state. Iterative model building and real space refinement were performed using COOT[62] and PHENIX.refine[63]. The refinement statistics are summarized in Table 1.

The ligand (L-lysine) was modeled in the electron density using the GemSpot workflow[64]. This involves molecular docking with Glide[65], followed by real-space refinement with Phenix/OPLS3e[66]. To enhance confidence in ligand placement, the final lysine-bound PaLysP was aligned in PyMOL (Schrödinger LLC, NY, USA) to arginine-bound GkApcT (PDB ID: 6F34) and arginine-bound EcAdiC (PDB ID: 3L1L). Satisfyingly, the ligand's backbone and side chain in all three complexes aligned.

### Nanobody generation, expression and purification

Nanobodies (Nb5753, Nb5755, Nb5758, Nb5760, Nb5761, Nb5764, and Nb5765) against LysP were generated using previously published protocols[67]. In brief, one llama (Lama glama) was six times immunized with LysP (1 mg/mL in DDM purification buffer). Four days after the final boost, blood was taken from the llama to isolate peripheral blood lymphocytes. RNA was purified from these lymphocytes and reverse transcribed by PCR to obtain the cDNA of the open reading frames coding for the nanobodies. The resulting library was cloned into the phage display vector pMESy4 bearing a C-terminal His$_6$ tag and a CaptureSelect sequence tag (Glu-Pro-Glu-Ala). Nanobodies were selected by biopanning. For this, LysP in detergent solution (150 mM NaCl, 20 mM Tris-HCL pH 7.5 and 0.03% (w/v) DDM) was solid phase coated directly on enzyme-linked immunosorbent assay (ELISA) plates. LysP specific phage were recovered by limited trypsinization, and after two rounds of selection, periplasmic extracts were made and analysed using ELISA screens. Nanobodies were expressed in E. coli Bl21 (DE3) for subsequent purification from the bacterial periplasm. After Ni$^{2+}$-NTA (Qiagen) affinity purification, nanobodies were further purified by size exclusion chromatography in buffer containing 20 mM Tris-HCl pH 7.5 and 150 mM NaCl. The purity of the purified nanobodies were analysed by SDS PAGE. The nanobodies were stored in the cold (4 °C) until use.

### Green fluorescent protein-thermal shift (GFP-TS) assay

The GFP-TS assay was performed as previously described[68,69]. Briefly, membranes containing the LysP-GFP fusion were thawed from −80 °C storage (3.5 mg/mL in 150 mM NaCl, 20 mM Tris-HCL pH 7.5 and 1% (w/v) DDM) were solubilized with or without 1 μM nanobody Nb5755 for 1 h at 4 °C with mild agitation. After solubilization, β-OG was added to both samples to a final concentration of 1%(w/v) and 120 μL aliquots of the solubilized membranes were incubated in 200 μL PCR tubes for 10 min at 4, 20, 30, 35, 40, 45, 50, 65, 70, or 100 °C using a T100$^{TM}$ Thermal Cycler (BIO-RAD) without mixing. The heated samples were then centrifuged at 18,000 × g for 30 min at 4 °C using a Microfuge® 18 Centrifuge, Beckman Coulter. 90 μL aliquots of the supernatant were transferred into a black clear bottom 96-well plate (Nunc) and the GFP fluorescence (excitation 485 nm, emission 512 nm) measured using a SpectraMax Germini EM microplate reader (Molecular Devices). Melting temperatures ($T_m$) were determined by plotting normalized fluorescence intensity versus temperature and fitting the data to a sigmoidal Boltzmann curve using GraphPad Prism, with $T_m$ defined as the midpoint of the unfolding transition.

### Molecular docking

Modeling studies were performed using programs of the Schrödinger Small-Molecule Drug Discovery Suite 2021-4 (Schrödinger LLC, NY, USA). The 2D structures of L-lysine, L-thialysine, and L-arginine were converted to geometry optimized 3D structures of their predominant ionization states (at pH 7.4) using LigPrep (Schrödinger LLC, NY, USA). Next, the cryo-EM structure of LysP in complex with L-Lys was prepared with the Protein PrepWizard[70]. This involves adding missing hydrogen atoms, adjusting the ionization state of polar amino acids at neutral pH while adjusting bond orders and formal charges of the ligand, optimizing the H-bond network of the protein−ligand complex,

and, finally, energetically minimizing the complex using the Optimized Potentials for Liquid Simulations (OPLS)−4 force field[71]. The geometric center of the bound L-lysine molecule was considered as the grid centroid, and flexible ligand docking was carried out using Glide with the SP scoring function[65].

## Normal mode analysis

The structural dynamics of LysP were analyzed using Normal Mode Analysis (NMA), a powerful method that samples the different conformational landscapes of proteins by employing topological constraints within the protein structure and empirical force fields[72]. To generate various conformational states of LysP based on the inward-occluded structure obtained by cryo-EM in this study, NMA was performed using the web server ELNemo[72]. The slow modes obtained from NMA provided insights into the hypothetical mode of action of LysP. These different conformational states were combined into a movie (Supplementary Movie 1) using PyMOL, illustrating the dynamic behavior of LysP.

## Preparation of Figures

Graph preparation and statistical analysis were performed in GraphPad Prism 10 (for macOS). Sequence alignments were generated using Clustal Omega online software[73]. Figures with models and EM density maps were prepared in PyMOL (the PyMOL Molecular Graphics System, Version 3.0.1, Schrödinger)[74], UCSF Chimera[75] and ChimeraX[76].

## Reporting summary

Further information on research design is available in the Nature Portfolio Reporting Summary linked to this article.

## Data availability

The structural model of the LysP-CA5755-L-lysine complex has been deposited in the Protein Data Bank under accession code 9EYD. The cryo-EM maps were deposited in the Electron Microscopy Data Bank (EMDB) under accession number EMD-50053. Source data are provided with this paper. Materials are available from the corresponding author upon request. Source data are provided with this paper.

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

## Acknowledgements

This research was funded by the Wellcome Trust (grant 222999/Z/21/Z to E.N.), Google Award for Socially Impactful Project enabled by AlphaFold to E.N., the Wenner-Gren Foundations (grant GFOv2022-0003 to J.J.G.), the Swedish Research Council (grant 2022-02985 to J.J.G.) and the Science Foundation Ireland (grant 12/IA/1255 and 22/FFP-A/10278 to M.C.). P.S. and P.J.F.H. are grateful to the European Commission for support from the EDICT grant 201924. We acknowledge the use of the Cryo-EM Uppsala facility for grid preparation and screening, funded by the Department of Cell and Molecular Biology, the Disciplinary Domains of Science and Technology and of Medicine and Pharmacy at Uppsala University. We also acknowledge the Membrane Protein Laboratory at Diamond Light Source funded by Wellcome (223727/Z/21/Z). Diamond Light Source and the Research Complex at Harwell are both Instruct-ERIC centres. Cryo-EM data was collected at the Cryo-EM Swedish National Facility funded by the Knut and Alice Wallenberg, Family Erling Persson and Kempe Foundations, SciLifeLab, Stockholm University and Umeå University. We thank Daniel Larsson, and Marta Carroni for assistance with cryo-EM data collection. We also acknowledge the support and the use of resources of Instruct-ERIC, part of the European Strategy Forum on Research Infrastructures (ESFRI), and the Research Foundation - Flanders (FWO) for their support to the Nanobody discovery project. We thank Nele Buys for the technical assistance during Nanobody discovery.

## Author contributions

E.N. designed and supervised the project. E.N. performed protein expression, purification, functional assays and cryo-EM sample preparation and data collection, as well as atomic model building and refinement. D.B. and J.J.G. assisted with protein production and cryo-EM sample preparation, A.F.A.M. and D.B. performed docking studies, H.C. and A.Q. assisted with cryo-EM grid preparation, P.S. assisted with radioactivity assays. R.M. provided assistance with cryo-EM data processing. M.C. supervised the initial protein production, purification and biophysical characterization. P.J.F.H. supervised radioactivity assays. E.P. and J.S. performed nanobody generation. E.N. expressed and purified the mutants and A.S. performed the transport assays under the guidance of D.D. E.N. wrote the manuscript. All authors read and commented on the manuscript.

## Competing interests

The authors declare no competing interests.
