## [Transparent Peer Review file · Nature Communications]

Structural basis of specific lysine transport by *Pseudomonas aeruginosa* permease LysP

Corresponding Author: Dr. Emmanuel Nji

Version 1:

Reviewer comments:

Reviewer #1

(Remarks to the Author)

The article by Bicer et al. reports the cryo-EM structure of bacterial lysine-specific permease (LysP) from *Pseudomonas aeruginosa*. The study provides insights into LysP's overall structure, its interaction with the substrate L-lysine, and results from transport assays. Additionally, a comparative analysis with similar transporters is presented, leading to a proposed mechanism for substrate specificity and proton driven transport.

However, despite these contributions, the article does not meet the high-impact criteria typically expected for publication in Nature Communications. Specifically:

- **Figure Presentation:** Figures 2A-D detail the biochemical characterization of LysP with a nanobody. Traditionally, this type of data is reserved for extended data sections rather than the main text, where figures are highly valued. Additionally, there is some redundancy in the results section where the interactions between the nanobody and LysP are discussed.
- **Electron Density:** Figure 4c and Extended Figure 3a present the electron density for the LysP binding site. The density is described as sausage-shaped, suggesting that the substrate could fit in multiple orientations. The manuscript does not clearly explain how the authors determined the specific position of L-lysine.
- **Functional Assays:** Given the functional assay data, it would be expected that the authors would perform mutational analysis to identify essential residues in the active site. Such experiments could provide deeper insights into the function of LysP and aid in determining the orientation of lysine in the binding site.
- **Docking Studies:** The docking studies with thialysine offer limited insight into why it inhibits transport. Although it fits well in the binding pocket, the manuscript does not clarify why this binding prevents transport.
- **Selectivity Data:** The analysis of L-lysine selectivity is based on comparisons with other structures and docking studies but lacks direct experimental data to support the proposed selectivity by LysP. The impact of the manuscript would be significantly strengthened by demonstrating lysine selectivity through functional analysis.
- **Mechanistic Insights:** In the proposed mechanism (final section of the manuscript), Glu112 is introduced as essential for the protonation step during transport. However, this claim is not supported by any pieces of data for the *Pseudomonas aeruginosa* LysP.
- **Antibiotic Target Potential:** The potential for targeting LysP as an antibiotic is proposed but not experimentally validated. The hypothesis that targeting LysP could inhibit the growth of *Pseudomonas aeruginosa* lacks supporting experimental evidence. Moreover, reference 50 that is presented as evidence for the potential of bacterial growth inhibition by thialysine shows that a different protein, lysyl-tRNA synthetase, is the target of thialysine.

Overall, while the manuscript provides new structural and functional data, it does not meet the caliber required for publication in Nature Communications.

Reviewer #2

(Remarks to the Author)

The article by Bicer et al. report a structure and function study on the lysine-specific antiporter LysP from *Pseudomonas aeruginosa*. A 3.7 Å cryo-EM structure of LysP bound to a stabilizing nanobody and in

complex with L-lysine is reported together with biochemical results establishing that LysP is specific for L-lysine, inhibited by L-4-thialysine, and requires a pH gradient to operate. They find that their structure is of the inward occluded state and they interpret their biochemical results using the cryo-EM structure with the help of molecular docking. Finally, they attempt to establish plausible structures of LysP in its different conformational states using normal mode analysis.

Overall the cryo-EM, modeling, and biochemical experiments appear to have been done well. The normal mode analysis may also have been done well, but there is not enough detail in the methods section at least to be sure (this will be described below). I greatly appreciate the authors sharing their structure and cryo-EM map; this allows me to do a much more thorough job of reviewing their work than I could do otherwise. I do have a number of concerns, a couple of them major, about the manuscript that need to be addressed however.

Major concern:

1. The authors claim that "The binding of L-lysine and the protonation of Glu112 on TM3 result in the movement of TM6 towards TM3, forming a strong cation- π interaction between Phe215 on TM6 and the protonated Glu112 of TM3." In order to form a cation- π interaction, Glu112 would need to be doubly protonated. Although this is not impossible, it is also not a very frequent occurrence, and I would not expect it to occur often enough to account for the frequency of lysine transport. (The pKa for the second protonation of a carboxylic acid is typically somewhere between -8 and 0 depending on circumstances.) A second problem is that the geometry of the side-chains, at least in the structure provided, is not correct for a cation- π interaction to form in the first place.

I would therefore suggest that the authors reconsider their proposed mechanism for triggering the conformational change. A mechanism that makes use of a water molecule, like that of GkApcT in reference 40, would probably be preferable (though the lack of an opposing negatively charged residue corresponding to the Asp237 in GkApcT likely means that the details of the mechanism would not be the same). Alternatively, they could either provide additional experimental support for this mechanism or identify an analogous, experimentally well-supported case of a solvent-exposed, doubly-protonated Glu or Asp residue driving a similar conformational change in the literature.

Moderate concerns:

1. The bound lysine is described as making "strong cation- π " interactions with Trp105. I agree that there is likely to be a cation- π interaction, but I am not sure it can be categorized as "strong" (that is, stronger than an average hydrogen-bond):

a) in the structure provided, the C ϵ atom looks like it's more likely to be part of the group making the cation- π interaction than the NH $_3^+$ group; this group has only about half the partial charge of the NH $_3^+$ group

b) the distance from the lysine C ϵ atom to the Trp105 6-membered ring is 4.9 Å, which is close enough to be a reasonable interaction, but too far to be optimum

c) the side-chain of Asn104 is positioned such that it may slightly screen the cation- π interaction without actually blocking it.

In contrast, the hydrogen-bond between the bound lysine N ζ atom and the side-chain oxygen of Ser377 is only 2.2 Å, making it a short, very strong (potentially low-barrier) hydrogen-bond. While I agree that Trp105 is likely to play an important role, it may be that Ser377 is also important for LysP lysine specificity (this residue is also conserved in Figure S1).

2. The authors used normal mode analysis (NMA) to generate the different conformational states of LysP that presumably correspond to the different conformational states shown in Figure 6b. These states were then used to construct supplementary movie 1. It is not clear from the manuscript, but it appears that the authors also used the results of NMA to develop their hypothesis that the interaction between Glu112 and Phe215 is important for channel opening. The only remark that the authors make on the subject is in the Methods section: "The slow modes obtained from NMA provided insights into the hypothetical mode of action of LysP."

There are two aspects that need consideration here:

a) There is more than one way to generate different conformational states using NMA. Given what was written, I assume the authors followed the simplest and easiest method: they calculated the normal modes using their cryo-EM structure, used those modes to create different conformations, and then selected those conformations that looked most similar to the conformational states they desired. This is legitimate and may be quite acceptable in the absence of additional information. The authors should, at the least, indicate which modes were used to infer which conformational states, however.

(One caution when using this procedure is that it is often tempting to examine only the first few normal modes and neglect the higher frequency one, but it has often been found that large conformational changes involve a mixture of both lower and higher frequency modes. As a practical matter, this means that conformations drawn from only one mode may broadly capture the large-scale changes that take place but miss important details.)

b) If the authors did use NMA to develop their hypothesis, then they should note this explicitly in the manuscript (e.g. "The structure of LysP was analyzed with the help of NMA to produce plausible conformations of LysP corresponding to the outward-open, outward-occluded, occluded, and inward-open conformations of GkApcT and EcAdiC. This analysis suggests that...")

Minor concerns:

1. TM10 & TM11 are not linked by a β -hairpin. DSSP characterizes it as a hydrogen-bonded turn followed by a bend.

2. The docking studies in the Methods section describe only L-lysine and L-arginine. Was L-thialysine docked in the same way?

3. Lines 219–220: although a T-stacked Arg–Trp interaction is less favorable than a π -stacked one, it is not really "unfavorable" considered in isolation. The conclusion that Arg is just too large and bulky to fit into the lys binding pocket is the correct one.

4. In all figures where error bars are shown, they are reported as "mean \pm s.e.m. of the fit." Although it's more normal to report standard deviation when describing a data set, especially when there are only a few measurements, there's nothing inherently wrong with reporting SEM. What does confuse me is what is meant by the "s.e.m. of the fit". When there is a fitted curve, does it mean that the error bars are reporting the residuals? Or do they report the expected confidence interval? But if this were the case then they should be symmetric about the fitted line and they normally aren't. It is especially confusing when this is applied to a bar graph. What exactly is being fit in Figures 1c and S7e and S7f?

(On the subject of bar graphs, I would like to commend the authors for including the data points in their bar graphs. This is an excellent practice that should be much more common than it currently is. I would suggest making the data spots a lighter color, though.)

5. In Figure 4, it would help readability if the ligplot diagram in panel b was in the same general orientation as the figures in the other three panels.
6. In figure 5d–g, the current version of PyMOL allows measurements to ring centers. This is probably preferable when illustrating cation- π interactions.
7. In Figure 6, the schematic pictures in panel b are in a reversed orientation to those in panel a (that is, helices 3 and 10 are on the left in panel a but on the right in panel b). It would be better if both the schematic and the model were in the same orientation.
8. Table 1: You did have an initial model, even if it wasn't from the PDB.
9. Figure S5. The title is not correct. It should be simply “The LysP-Nb5755 interaction.” It would also be beneficial to explicitly note what is colored in the figure caption.
10. Figure S6. Panel a illustrates the cation- π interaction between thialysine and Trp105, not L-lysine.
11. In the movie, it would be good if the individual conformational states were labeled so the viewer knows which one is being shown. Also, while Phe215 is shown, Glu112 is not.
12. Something that I'm just curious about: Is the cadaverine exported from the cell in its deprotonated form so that it absorbs an extra 2 H^+ from the extracellular environment, or does this mechanism reduce the external H^+ concentration only through the decarboxylation of lysine in the same way that AdiA and AdiC do through the decarboxylation of arginine?

Reviewer #3

(Remarks to the Author)

Reviewer #4

(Remarks to the Author)

Bicer and colleagues reported the structure of the *Pseudomonas aeruginosa* lysine-specific permease (LysP) determined using single-particle cryo-EM at a resolution of 3.7 Å. The structure captured LysP in complex with L-lysine and a nanobody, which was used to stabilize the complex for cryo-EM analysis. The cryo-EM map is of reasonable quality, with a resolution of 3.7 Å. Data was collected using excellent equipment and analyzed with standard software (CryoSPARC). The model was built using Coot and refined following standard Phenix refinement protocols, with statistics falling within the 50th percentile of cryo-EM structures.

While the data presented in the publication is valuable, there is an opportunity to enhance the clarity and flow of the manuscript. Improving the organization of the figures to follow a more logical sequence would greatly help guide the reader through the findings. Currently, the results section refers back and forth between figures in a way that disrupts the flow. For example, it begins with Extended Data Fig. 4, then jumps to Fig. 2, followed by Fig. 1b, then Fig. 3, Fig. 6b, and back to Fig. 2. A more linear progression in the figure references would make the manuscript easier to follow and improve overall clarity. The quality of the presentation affects not only its aesthetic appeal and logical flow but also the manuscript's ability to clearly present and demonstrate key findings to support the conclusions. In particular, several claims are made but are not fully illustrated or demonstrated in the figures, such as:

- The claim that the structure represents an inward-occluded state, as mentioned in Fig. 6b (in a cartoon representation), is not clearly shown. A figure that highlights the presence of a tunnel or cavity in the structure—closed on one side and open on the other—would better visualize and support this structural state.
- The claim that L-lysine is coordinated by hydrophobic stacking with Phe215 and cation- π interactions with Trp105 is not clearly demonstrated in Fig. 4a, b, and c due to a lack of labels in the images. Similarly, the coordination of the ϵ -amino group through hydrogen bonds is unclear.
- Glu112 protonation is proposed to induce movement of TM6, but this is not explicitly shown in any figure, aside from the cartoon in Fig. 6b.
- Trp105 is suggested to act as a gate, yet this is also not clearly shown.

Importantly, the authors should demonstrate that the electron density maps are well-defined in these areas and provide sufficient evidence to substantiate these claims. Moreover, given that the authors have established an assay to measure LysP activity, it is surprising that no mutagenesis studies have been conducted on Trp105, Phe215, and Glu112. Performing mutagenesis on these residues would significantly strengthen the work presented in this manuscript.

Version 2:

Reviewer comments:

Reviewer #1

(Remarks to the Author)

The authors have addressed all of my comments, and the manuscript has been revised accordingly. Notably, the updated version incorporates a new analysis showing the critical role of Lys162 in transport. I would, however, caution that although Lys162 appears essential, its role may not necessarily be linked to proton coupling, though I understand the reasoning behind this interpretation.

Reviewer #2

(Remarks to the Author)

The authors have properly addressed all the points I raised in my original review. Overall, I am pleased with the revisions.

In reading the new version, I did find a few minor points that should probably be addressed before the article is published, however:

1. Figure 1a has three small mistakes:

a. The stick figure for lysine that has been added is in its neutral form rather than the expected zwitterionic form. This is particularly strange since the side-chain is shown in its protonated form. This should be fixed.

b. The label for CadC has somehow dropped out.

c. One of the products of CadA is CO₂ (C-zero-two), this should be CO₂ (C-O-two).

2. In Figure 7b, ARG162 should be LYS162. Also, it is an arginine side-chain that is shown in the illustration rather than a lysine.

3. Page 7, line 196 and page 8, lines 236–237: In the revised structure the distance between the N ζ atom of the bound lysine and the side-chain oxygen atom of Ser-377 is 3.0 Å. This distance is in the middle of the normal hydrogen-bonding range (typically taken to be 2.8–3.2 Å), so it is no longer appropriate to call it a short, very strong, potentially low-barrier hydrogen bond. (This does not change the authors' conclusions that residues Ser-377 and Asn-104 are important for determining the specificity of LysP.)

The following three suggestions only relate to the arrangement of the manuscript rather than its technical content:

4. Page 6, lines 169–172: The description of the Lys-162 results interrupts the description of the lysine binding site. Lys-162 is not part of the lysine binding site and does not appear in any of the figures showing the bound lysine (it doesn't appear until Figure 7). I would suggest moving this part to a separate subsection near the end of the Results (this might require reworking and expanding it slightly).

5. Pages 7–8, lines 199–207. This part should really go into the Discussion someplace, not the results (Perhaps following lines 252–253).

6. A very minor point: in the first subsection of the Results, the pH gradients are described using phrases like "pH 7 in and pH 4 out", etc. I would suggest changing the words "in" and "out" to "inside" and "outside" in this context.

Reviewer #3

(Remarks to the Author)

All of my comments were addressed and the manuscript was modified accordingly.

The updated manuscript includes a new analysis that found Lys162 to be essential for transport. The authors claim that "In addition, we identified Lys162 as essential for proton coupling, as its substitution with alanine abolished L-lysine transport (Fig. 4e). Lys162 is homologous to Lys158 in *Methanocaldococcus jannaschii* ApcT (Fig. 4f), previously shown to mediate proton coupling, and corresponds to the second sodium-binding site in Leu". This interpretation of "Lys162 as essential for proton coupling" has not been demonstrated by the authors since many other sites in the protein could be mutated to alanine and lose activity due to a number of reasons, not necessarily proton coupling. This claim of proton coupling should be toned down and the mechanism at line 268 should be modified, unless experimentally proven by comparing the pH dependence of uptake of wild-type protein to that of a K162R mutant.

Reviewer #4

(Remarks to the Author)

Since the last version of the manuscript, the authors have added a substantial number of mutagenesis experiments to support their study. They have also carefully considered the comments from other reviewers and have been very meticulous in their wording in order to avoid overinterpretation or misleading interpretation.

I believe they have appropriately addressed all comments, and I support publication of the manuscript in the Journal. I believe the quality of the work is appropriate to this Journal.

We sincerely thank the reviewers for their valuable feedback, which has significantly enhanced the quality of the manuscript. We have addressed their concerns in detail below.

Reviewer #1 (Remarks to the Author)

The article by Bicer et al. reports the cryo-EM structure of bacterial lysine-specific permease (LysP) from *Pseudomonas aeruginosa*. The study provides insights into LysP's overall structure, its interaction with the substrate L-lysine, and results from transport assays. Additionally, a comparative analysis with similar transporters is presented, leading to a proposed mechanism for substrate specificity and proton driven transport.

However, despite these contributions, the article does not meet the high-impact criteria typically expected for publication in Nature Communications. Specifically:

- Figure Presentation: Figures 2A-D detail the biochemical characterization of LysP with a nanobody. Traditionally, this type of data is reserved for extended data sections rather than the main text, where figures are highly valued. Additionally, there is some redundancy in the results section where the interactions between the nanobody and LysP are discussed.

We thank the Reviewer for bringing this to our attention. In response, we have moved Figures 2A–D, which detail the biochemical characterization of the LysP–nanobody interaction, to the Supplementary Information (now Supplementary Figures 4A-D). Additionally, we have revised the corresponding text in the Results section to eliminate redundancy and streamline the discussion of the LysP-nanobody interactions (Lines 214-227).

- Electron Density: Figure 4c and Extended Figure 3a present the electron density for the LysP binding site. The density is described as sausage-shaped, suggesting that the substrate could fit in multiple orientations. The manuscript does not clearly explain how the authors determined the specific position of L-lysine.

The following description has now been added to the manuscript in the Methods section (Lines 498 – 503) “The ligand was modeled in the electron density using the GemSpot workflow. This involves molecular docking with Glide, followed by real-space refinement with Phenix/OPLS3e. To enhance confidence in ligand placement, the final lysine-bound PaLysP was aligned in PyMOL (Schrödinger LLC, NY, USA) to arginine-bound GkApcT (PDB ID: 6F34) and arginine-bound EcAdiC (PDB ID: 3L1L). Satisfyingly, the ligand’s backbone and side chain in all three complexes aligned.”

- **Functional Assays:** Given the functional assay data, it would be expected that the authors would perform mutational analysis to identify essential residues in the active site. Such experiments could provide deeper insights into the function of LysP and aid in determining the orientation of lysine in the binding site.

We thank the Reviewer for this valuable feedback. We performed functional assays targeting residues within the LysP binding site, which further validated key residues essential for L-lysine binding. Indeed, these mutational analyses provided additional insights into the functional role of specific residues and support the proposed orientation of L-lysine within the binding pocket. We believe these results complement our structural findings and strengthen the proposed LysP's transport mechanism.

$[^3\text{H}]$ -Lysine transport assays of wild-type LysP and binding site mutants (W105Y, W105A, E112A, F215A and N104A).

- **Docking Studies:** The docking studies with thialysine offer limited insight into why it inhibits transport. Although it fits well in the binding pocket, the manuscript does not clarify why this binding prevents transport.

We thank the Reviewer for this thoughtful observation. The primary purpose of our docking studies with thialysine was to assist in validating the proposed L-lysine binding site and the L-lysine orientation. These results, now complemented by recent transport assays targeting specific binding site residues, provide strong support for our structural interpretation. Thialysine is a close structural analogue of L-lysine, differing only by the substitution of the γ -methylene group with a sulfur atom (as shown in the figure below: A) L-lysine; B) thialysine). This subtle change preserves both binding and transport compatibility with LysP. We have revised the manuscript to clarify the structural differences between thialysine and lysine, and to explain how this substitution results in competitive inhibition of lysine transport.

Lysine Thialysine

- **Selectivity Data:** The analysis of L-lysine selectivity is based on comparisons with other structures and docking studies but lacks direct experimental data to support the proposed selectivity by LysP. The impact

of the manuscript would be significantly strengthened by demonstrating lysine selectivity through functional analysis.

We thank the Reviewer for this valuable comment. In response, we have now performed mutagenesis on Trp105, Phe215, Glu112, and other residues. The results from these experiments have provided important functional insights and have significantly strengthened the conclusions presented in the manuscript. We have included the new data in the revised Fig. 4 and updated the Results and Discussion sections accordingly.

- Mechanistic Insights: In the proposed mechanism (final section of the manuscript), Glu112 is introduced as essential for the protonation step during transport. However, this claim is not supported by any pieces of data for the *Pseudomonas aeruginosa* LysP.

We have further refined our structure and found that Glu112 forms a direct hydrogen bond with the lysine ligand. Mutation of this residue to alanine abolished lysine transport, corroborating its essential role in substrate recognition rather than proton coupling (Fig. 4d). In contrast, the K162 residue on TM5 does not make direct contact with the lysine ligand but is homologous to K158 in MjApcT, which plays a critical role in proton coupling and corresponds to the second sodium-binding site in LeuT. Similarly, the K162A mutation in LysP abolished lysine transport, indicating that this conserved lysine is crucial for proton coupling in the transport mechanism.

[³H]-Lysine transport assays of wild-type LysP and proton coupling mutant (K162A).

- Antibiotic Target Potential: The potential for targeting LysP as an antibiotic is proposed but not experimentally validated. The hypothesis that targeting LysP could inhibit the growth of *Pseudomonas aeruginosa* lacks supporting experimental evidence. Moreover, reference 50 that is presented as evidence for the potential of bacterial growth inhibition by thialysine shows that a different protein, lysyl-tRNA synthetase, is the target of thialysine.

We thank the Reviewer for this important observation. We agree that our hypothesis regarding the potential of LysP as an antibiotic target is speculative and not yet supported by direct experimental validation. Our intention was to highlight a possible therapeutic opportunity that might be exploited based on the structural and functional characterization of LysP, particularly its specificity for L-lysine and transport inhibition by L-4-thialysine.

We acknowledge that reference 50 demonstrates inhibition of bacterial growth by targeting lysyl-tRNA synthetase, not LysP. We have revised the manuscript to clarify this distinction and have removed any

implication that thialysine directly targets LysP. Instead, we now frame the antibiotic potential of targeting LysP as a hypothesis that warrants further investigation, including assessing the impact of genetic or chemical inhibition of LysP on *P. aeruginosa* growth under lysine-limited conditions.

Overall, while the manuscript provides new structural and functional data, it does not meet the caliber required for publication in Nature Communications.

Reviewer #2 (Remarks to the Author)

The article by Bicer et al. report a structure and function study on the lysine-specific antiporter LysP from *Pseudomonas aeruginosa*. A 3.7 Å cryo-EM structure of LysP bound to a stabilizing nanobody and in complex with L-lysine is reported together with biochemical results establishing that LysP is specific for L-lysine, inhibited by L-4-thialysine, and requires a pH gradient to operate. They find that their structure is of the inward occluded state and they interpret their biochemical results using the cryo-EM structure with the help of molecular docking. Finally, they attempt to establish plausible structures of LysP in its different conformational states using normal mode analysis.

Overall the cryo-EM, modeling, and biochemical experiments appear to have been done well. The normal mode analysis may also have been done well, but there is not enough detail in the methods section at least to be sure (this will be described below). I greatly appreciate the authors sharing their structure and cryo-EM map; this allows me to do a much more thorough job of reviewing their work than I could do otherwise. I do have a number of concerns, a couple of them major, about the manuscript that need to be addressed however.

Major concern:

1. The authors claim that “The binding of L-lysine and the protonation of Glu112 on TM3 result in the movement of TM6 towards TM3, forming a strong cation- π interaction between Phe215 on TM6 and the protonated Glu112 of TM3.” In order to form a cation- π interaction, Glu112 would need to be doubly protonated. Although this is not impossible, it is also not a very frequent occurrence, and I would not expect it to occur often enough to account for the frequency of lysine transport. (The pKa for the second protonation of a carboxylic acid is typically somewhere between -8 and 0 depending on circumstances.) A second problem is that the geometry of the side-chains, at least in the structure provided, is not correct for a cation- π interaction to form in the first place.

I would therefore suggest that the authors reconsider their proposed mechanism for triggering the conformational change. A mechanism that makes use of a water molecule, like that of GkApcT in reference 40, would probably be preferable (though the lack of an opposing negatively charged residue corresponding to the Asp237 in GkApcT likely means that the details of the mechanism would not be the same).

Alternatively, they could either provide additional experimental support for this mechanism or identify an

analogous, experimentally well-supported case of a solvent-exposed, doubly-protonated Glu or Asp residue driving a similar conformational change in the literature.

We thank the Reviewer for their detailed and thoughtful critique regarding the proposed role of Glu112 and its interaction with Phe215. We agree that the likelihood of Glu112 being doubly protonated is low, and that the geometry observed in our structure does not support a classical cation- π interaction.

Further analysis of the cryo-EM density and ligand-binding site reveals that Glu112 is positioned within hydrogen-bonding distance (approximately 3.6 Å) of the α -carboxyl group of the bound lysine ligand (Fig. 4b & 5d), suggesting that it likely participates in substrate coordination through hydrogen bonding rather than direct protonation. This interaction may help stabilize the ligand within the binding pocket and facilitate the subsequent conformational transition.

In contrast, evidence from the related *Methanocaldococcus jannaschii* broad spectrum amino acid transporter (MjApcT) indicates that proton coupling occurs via a conserved lysine (K158), homologous to K162 in *Pseudomonas aeruginosa* LysP, which aligns with the second sodium-binding site in LeuT. Our mutagenesis data show that substitution of K162 with Ala abolishes lysine transport, strongly supporting its essential role in proton coupling.

Accordingly, we have revised the manuscript to reflect this refined mechanistic interpretation, removing the previous reference to a cation- π interaction and highlighting instead the role of Glu112 in hydrogen bonding and K162 in proton coupling (Fig 7a & b).

a. Sequence alignment of LysP with other cationic amino acid transporters, highlighting conserved lysine residues that correspond to the second sodium-binding site (Na₂) in LeuT. b. [³H]-Lysine transport assays of wild-type LysP and proton coupling mutant (K162A).

Moderate concerns:

1. The bound lysine is described as making “strong cation- π ” interactions with Trp105. I agree that there is likely to be a cation- π interaction, but I am not sure it can be categorized as “strong” (that is, stronger than an average hydrogen-bond):

We agree with the Reviewer. The word “strong” has been removed.

a) in the structure provided, the C ϵ atom looks like it's more likely to be part of the group making the cation- π interaction than the NH₃⁺ group; this group has only about half the partial charge of the NH₃⁺

group

The structure of the lysine-bound LysP complex has been further refined to resolve this discrepancy.

b) the distance from the lysine C ϵ atom to the Trp105 6-membered ring is 4.9 Å, which is close enough to be a reasonable interaction, but too far to be optimum

We agree with the Reviewer. In the revised structure, the distance between lysine's C ϵ atom to the Trp105's 6-membered ring is 5.6 Å, excluding the likelihood of an optimal interaction.

c) the side-chain of Asn104 is positioned such that it may slightly screen the cation- π interaction without actually blocking it.

We agree with the Reviewer.

In contrast, the hydrogen-bond between the bound lysine N ζ atom and the side-chain oxygen of Ser377 is only 2.2 Å, making it a short, very strong (potentially low-barrier) hydrogen-bond. While I agree that Trp105 is likely to play an important role, it may be that Ser377 is also important for LysP lysine specificity (this residue is also conserved in Figure S1).

We agree with the Reviewer. The following sentence has been added to the Results section of the revised manuscript. "Given that Ser377 is not conserved among cationic amino transporters, we hypothesize that this residue also plays a role in the specificity of LysP toward L-lysine." (Line 197-198)

2. The authors used normal mode analysis (NMA) to generate the different conformational states of LysP that presumably correspond to the different conformational states shown in Figure 6b. These states were then used to construct supplementary movie 1. It is not clear from the manuscript, but it appears that the authors also used the results of NMA to develop their hypothesis that the interaction between Glu112 and Phe215 is important for channel opening. The only remark that the authors make on the subject is in the Methods section: "The slow modes obtained from NMA provided insights into the hypothetical mode of action of LysP."

There are two aspects that need consideration here:

a) There is more than one way to generate different conformational states using NMA. Given what was written, I assume the authors followed the simplest and easiest method: they calculated the normal modes using their cryo-EM structure, used those modes to create different conformations, and then selected those conformations that looked most similar to the conformational states they desired. This is legitimate and may be quite acceptable in the absence of additional information. The authors should, at the least, indicate which modes were used to infer which conformational states, however. (One caution when using this procedure is that it is often tempting to examine only the first few normal modes and neglect the higher frequency one, but it has often been found that large conformational changes involve a mixture of both lower and higher frequency modes. As a practical matter, this means that conformations drawn from only one mode may broadly capture the large-scale changes that take place but miss important details.)

b) If the authors did use NMA to develop their hypothesis, then they should note this explicitly in the manuscript

(e.g. “The structure of LysP was analyzed with the help of NMA to produce plausible conformations of LysP corresponding to the outward-open, outward-occluded, occluded, and inward-open conformations of GkApcT and EcAdiC. This analysis suggests that...”)

We thank the Reviewer for bringing these important points about NMA to our attention. We have included the suggested modification in the revised manuscript as follows:

The structure of LysP was analyzed with the help of normal mode analysis (NMA) to produce plausible conformations of LysP corresponding to the outward-open, outward-occluded, occluded, and inward-open conformations of LysP. (line 254-257)

Minor concerns:

1. TM10 & TM11 are not linked by a β -hairpin. DSSP characterizes it as a hydrogen-bonded turn followed by a bend.

We thank the Reviewer for pointing out this error. We have revised the manuscript to reflect this change.

2. The docking studies in the Methods section describe only L-lysine and L-arginine. Was L-thialysine docked in the same way?

We thank the Reviewer for pointing out this omission. Indeed, thialysine was docked in the exact same way. The Methods section was amended accordingly (Line 570-574).

3. Lines 219–220: although a T-stacked Arg–Trp interaction is less favorable than a π stacked one, it is not really “unfavorable” considered in isolation. The conclusion that Arg is just too large and bulky to fit into the lys binding pocket is the correct one.

We agree with the Reviewer’s interpretation that the primary reason for Arg exclusion is its larger size, which prevents it from fitting properly into the lysine binding pocket.

To support this conclusion, we refer to the W105Y mutation, which enlarges the binding site and results in a 70% reduction in lysine transport activity (Fig. 4d). This finding suggests that optimal binding and transport require a more confined pocket, as provided by Trp105, to accommodate lysine specifically.

We have revised the manuscript accordingly to reflect this conclusion and removed the texts suggesting that T-stacked Arg–Trp interactions are intrinsically unfavorable.

4. In all figures where error bars are shown, they are reported as “mean \pm s.e.m. of the fit.” Although it's more normal to report standard deviation when describing a data set, especially when there are only a

few measurements, there's nothing inherently wrong with reporting SEM. What does confuse me is what is meant by the "s.e.m. of the fit". When there is a fitted curve, does it mean that the error bars are reporting the residuals? Or do they report the expected confidence interval? But if this were the case then they should be symmetric about the fitted line and they normally aren't. It is especially confusing when this is applied to a bar graph. What exactly is being fit in Figures 1c and S7e and S7f?

We agree that the original description, 'mean \pm s.e.m. of the fit', is unclear and potentially misleading. To address this, we have revised all relevant figure legends for clarity to now read: 'Values represent the mean \pm s.e.m. of three independent experiments.'

(On the subject of bar graphs, I would like to commend the authors for including the data points in their bar graphs. This is an excellent practice that should be much more common than it currently is. I would suggest making the data spots a lighter color, though.)

We thank the Reviewer for the encouraging feedback regarding the inclusion of individual data points in our bar graphs. We have implemented the Reviewer's suggestion to use lighter colors for data points to enhance clarity.

In response, we have revised all relevant figures by changing the data points to a lighter color, improving visual distinction without compromising their visibility.

We believe this adjustment enhances the readability of our figures and thank the Reviewer once again for their helpful input.

5. In Figure 4, it would help readability if the ligplot diagram in panel b was in the same general orientation as the figures in the other three panels.

We thank the Reviewer for this valuable suggestion. We have attempted to rotate the LigPlot diagram to match the orientation of the other panels. However, due to the constraints of the LigPlot software and the need to maintain accurate depiction of interactions, it was not possible to fully control the orientation without compromising clarity. Nonetheless, we have ensured that the current orientation represents the key ligand interactions clearly.

6. In figure 5d–g, the current version of PyMOL allows measurements to ring centers. This is probably preferable when illustrating cation- π interactions.

We thank the Reviewer for this useful suggestion. We have revised Figure 5d–g (now Fig. 5e) in which ring center measurements are used to illustrate cation- π interactions, as enabled in the latest version of PyMOL.

7. In Figure 6, the schematic pictures in panel b are in a reversed orientation to those in panel a (that is, helices 3 and 10 are on the left in panel a but on the right in panel b). It would be better if both the schematic and the model were in the same orientation.

We thank the Reviewer for pointing out the inconsistency. To improve clarity and maintain consistency across panels, we have revised Figure 6b to match the orientation shown in Figure 6a, ensuring helices 3 and 10 are positioned similarly in both panels. The updated figure has been included in the revised manuscript as Fig. 7a & b.

8. Table 1: You did have an initial model, even if it wasn't from the PDB.

The AlphaFold2 model used has now been included.

9. Figure S5. The title is not correct. It should be simply “The LysP-Nb5755 interaction.” It would also be beneficial to explicitly note what is colored in the figure caption.

We thank the Reviewer for pointing out this error. We have revised the text to read “The LysP–Nb5755 interaction.” Additionally, the figure reference has been updated from Fig. S5 to Fig. 6 (line 667).

10. Figure S6. Panel a illustrates the cation- π interaction between thialysine and Trp105, not L-lysine.

We thank the Reviewer for pointing out this error. This figure has been removed from the revised manuscript.

11. In the movie, it would be good if the individual conformational states were labeled so the viewer knows which one is being shown. Also, while Phe215 is shown, Glu112 is not.

Glu112 is now shown, and the individual conformational states have been labeled accordingly.

12. Something that I'm just curious about: Is the cadaverine exported from the cell in its deprotonated form so that it absorbs an extra 2 H^+ from the extracellular environment, or does this mechanism reduce the external H^+ concentration only through the decarboxylation of lysine in the same way that AdiA and AdiC do through the decarboxylation of arginine?

We thank the Reviewer for this insightful question. We agree that clarifying the mechanism of acid resistance is important for understanding the physiological relevance of lysine import via LysP.

To address this, we have clarified in the revised text that the primary acid-neutralizing effect of the CadA–CadB system arises from the consumption of a proton during the intracellular decarboxylation of lysine by CadA. The resulting product, cadaverine, is exported in its protonated form via the antiporter CadB in exchange for lysine. Therefore, similar to the AdiA–AdiC system (arginine decarboxylation), the acid resistance mechanism relies on proton consumption within the cytoplasm, not on proton removal from the extracellular space.

We have added a sentence to the Discussion section to make this mechanism and to clarify its analogy to the AdiA–AdiC system. This clarification should help the reader understand how lysine transport by LysP plays a critical upstream role in enabling CadA–CadB–mediated acid resistance.

Reviewer #3 (Remarks to the Author)

We thank the Reviewer for participating in this important initiative at Nature Communications.

Reviewer #4 (Remarks to the Author):

Bicer and colleagues reported the structure of the *Pseudomonas aeruginosa* lysine-specific permease (LysP) determined using single-particle cryo-EM at a resolution of 3.7 Å. The structure captured LysP in complex with L-lysine and a nanobody, which was used to stabilize the complex for cryo-EM analysis. The cryo-EM map is of reasonable quality, with a resolution of 3.7 Å. Data was collected using excellent equipment and analyzed with standard software (CryoSPARC). The model was built using Coot and refined following standard Phenix refinement protocols, with statistics falling within the 50th percentile of cryo-EM structures.

While the data presented in the publication is valuable, there is an opportunity to enhance the clarity and flow of the manuscript. Improving the organization of the figures to follow a more logical sequence would greatly help guide the reader through the findings. Currently, the results section refers back and forth between figures in a way that disrupts the flow. For example, it begins with Extended Data Fig. 4, then jumps to Fig. 2, followed by Fig. 1b, then Fig. 3, Fig. 6b, and back to Fig. 2. A more linear progression in the figure references would make the manuscript easier to follow and improve overall clarity.

The quality of the presentation affects not only its aesthetic appeal and logical flow but also the manuscript's ability to clearly present and demonstrate key findings to support the conclusions. In particular, several claims are made but are not fully illustrated or demonstrated in the figures, such as:

- The claim that the structure represents an inward-occluded state, as mentioned in Fig. 6b (in a cartoon representation), is not clearly shown. A figure that highlights the presence of a tunnel or cavity in the structure—closed on one side and open on the other—would better visualize and support this structural state.

We thank the Reviewer for the helpful feedback. In response, we have reorganized the figures to better align with the logical progression of the text. To more clearly support our assignment of the inward-occluded conformation of LysP, we now include a structural superposition of the experimentally determined cryo-EM model with the AlphaFold2-predicted outward-open conformation. This comparison demonstrates that the extracellular gate is closed in our inward-occluded structure, while the intracellular side remains partially open, with helices TM6b and TM1a shifted outward (Fig. 4a). These features are consistent with an inward-occluded state. We believe this addition more clearly illustrates and substantiates the conformational assignment described in the manuscript.

Superimposition of the AlphaFold2-predicted outward-open structure (grey) and the inward-occluded cryo-EM structure of LysP (green). Red arrow indicate the movement of TM1a and TM6b, suggesting conformational changes that facilitate L-lysine release into the cytoplasm.

- The claim that L-lysine is coordinated by hydrophobic stacking with Phe215 and cation- π interactions with Trp105 is not clearly demonstrated in Fig. 4a, b, and c due to a lack of labels in the images. Similarly, the coordination of the ϵ -amino group through hydrogen bonds is unclear.

We thank the Reviewer for highlighting these shortcomings in our original submission. We trust that the images in Fig. 4c make good these deficits.

- Glu112 protonation is proposed to induce movement of TM6, but this is not explicitly shown in any figure, aside from the cartoon in Fig. 6b.

We thank the Reviewer for the comment. Due to new experimental evidence (Fig. 4d & e) we have no longer make this statement.

- Trp105 is suggested to act as a gate, yet this is also not clearly shown.

We thank the Reviewer for highlighting the omission. This is now shown in Fig. 4a & c

Importantly, the authors should demonstrate that the electron density maps are well-defined in these areas and provide sufficient evidence to substantiate these claims. Moreover, given that the authors have established an assay to measure LysP activity, it is surprising that no mutagenesis studies have been conducted on Trp105, Phe215, and Glu112. Performing mutagenesis on these residues would significantly strengthen the work presented in this manuscript.

We thank the Reviewer for these important comments. In response to the Reviewers' suggestions, we have performed mutagenesis on the corresponding and additional residues (Fig. 4d & e). The Reviewer is referred to our response to Reviewer #1 regarding the selectivity data for further clarification.

Point-by-Point Response to Reviewers' Comments

Reviewer #1 (Remarks to the Author):

The authors have addressed all of my comments, and the manuscript has been revised accordingly. Notably, the updated version incorporates a new analysis showing the critical role of Lys162 in transport. I would, however, caution that although Lys162 appears essential, its role may not necessarily be linked to proton coupling, though I understand the reasoning behind this interpretation.

We thank the reviewer for the positive assessment of our revised manuscript. We appreciate the cautionary note regarding the interpretation of Lys162 in proton coupling. In response, we have revised the text to tone down our claim and clarify that while Lys162 is essential for transport, its specific role in proton coupling remains a possibility rather than a definitive conclusion. We now frame this as a hypothesis that will require further experimental validation.

Reviewer #2 (Remarks to the Author):

The authors have properly addressed all the points I raised in my original review. Overall, I am pleased with the revisions.

We thank the reviewer for the positive evaluation of our revised manuscript.

In reading the new version, I did find a few minor points that should probably be addressed before the article is published, however:

1. **Figure 1a has three small mistakes:**

- a. The stick figure for lysine that has been added is in its neutral form rather than the expected zwitterionic form. This is particularly strange since the side-chain is shown in its protonated form. This should be fixed.
- b. The label for CadC has somehow dropped out.
- c. One of the products of CadA is C0₂ (C-zero-two), this should be CO₂ (C-O-two).

We have carefully considered and fully addressed all of the reviewer's comments above.

2. In Figure 7b, ARG162 should be LYS162. Also, it is an arginine side-chain that is shown in the illustration rather than a lysine.

We have fully addressed this comment.

3. Page 7, line 196 and page 8, lines 236–237: In the revised structure the distance between the N ζ atom of the bound lysine and the side-chain oxygen atom of Ser-377 is 3.0 Å. This distance is in the middle of the normal hydrogen-bonding range (typically taken to be 2.8–3.2 Å), so it is no longer appropriate to call it a short, very strong, potentially low-barrier hydrogen bond. (This does not change the authors' conclusions that residues Ser-377 and Asn-104 are important for determining the specificity of LysP.)

We thank the reviewer for this clarification. In line with the suggestion, we have revised the text to remove the reference to a “short, very strong, potentially low-barrier hydrogen bond” and now describe the interaction more generally as a hydrogen bond (lines 202 - 204 and 232 - 235).

The following three suggestions only relate to the arrangement of the manuscript rather than its technical content:

4. Page 6, lines 169–172: The description of the Lys-162 results interrupts the description of the lysine binding site. Lys-162 is not part of the lysine binding site and does not appear in any of the figures showing the bound lysine (it doesn't appear until Figure 7). I would suggest moving this part to a separate subsection near the end of the Results (this might require reworking and expanding it slightly).

We agree with the reviewer and have now included a subsection hypothesizing Lys162 as a potential proton-coupling residue that warrants further investigation (lines 179 - 185).

5. Pages 7–8, lines 199–207. This part should really go into the Discussion someplace, not the results (Perhaps following lines 252–253).

This has now been moved to the Discussion section (lines 250 – 258).

6. A very minor point: in the first subsection of the Results, the pH gradients are described using phrases like “pH 7 in and pH 4 out”, etc. I would suggest changing the words “in” and “out” to “inside” and “outside” in this context.

We have addressed this point in lines 120–123 of the revised manuscript.

Reviewer #3 (Remarks to the Author):

All of my comments were addressed and the manuscript was modified accordingly. The updated manuscript includes a new analysis that found Lys162 to be essential for transport. The authors claim that *"In addition, we identified Lys162 as essential for proton coupling, as its substitution with alanine abolished L-lysine transport (Fig. 4e). Lys162 is homologous to Lys158 in Methanocaldococcus jannaschii ApcT (Fig. 4f), previously shown to mediate proton coupling, and corresponds to the second sodium-binding site in Leu"*. This interpretation of "Lys162 as essential for proton coupling" has not been demonstrated by the authors since many other sites in the protein could be mutated to alanine and lose activity due to a number of reasons, not necessarily proton coupling. This claim of proton coupling should be toned down and the mechanism at line 268 should be modified, unless experimentally proven by comparing the pH dependence of uptake of wild-type protein to that of a K162R mutant.

We thank the reviewer for the positive evaluation of our revised manuscript and for raising the important point about our interpretation of Lys162. As also noted by Reviewer #1, we acknowledge that while Lys162 is essential for transport, its direct role in proton coupling has not been demonstrated. In line with both reviewers' comments, we have revised the text to tone down this claim and now frame the possible involvement of Lys162 in proton coupling as a hypothesis that requires further experimental validation.

Reviewer #4 (Remarks to the Author):

Since the last version of the manuscript, the authors have added a substantial number of mutagenesis experiments to support their study. They have also carefully considered the comments from other reviewers and have been very meticulous in their wording in order to avoid overinterpretation or misleading interpretation.

I believe they have appropriately addressed all comments, and I support publication of the manuscript in the Journal. I believe the quality of the work is appropriate to this Journal.

We sincerely thank the reviewer for the positive feedback and for supporting publication of our manuscript.